# Edge-strand of BepA interacts with immature LptD on the β-barrel assembly machine to direct it to on- and off-pathways

**Ryoji Miyazaki†, Tetsuro Watanabe, Kohei Yoshitani, Yoshinori Akiyama***

Institute for Frontier Life and Medical Sciences, Kyoto University, Kyoto, Japan

**Abstract** The outer membrane (OM) of Gram-negative bacteria functions as a selective permeability barrier. *Escherichia coli* periplasmic Zn-metallopeptidase BepA contributes to the maintenance of OM integrity through its involvement in the biogenesis and degradation of LptD, a β-barrel protein component of the lipopolysaccharide translocon. BepA either promotes the maturation of LptD when it is on the normal assembly pathway (on-pathway) or degrades it when its assembly is compromised (off-pathway). BepA performs these functions probably on the β-barrel assembly machinery (BAM) complex. However, how BepA recognizes and directs an immature LptD to different pathways remains unclear. Here, we explored the interactions among BepA, LptD, and the BAM complex. We found that the interaction of the BepA edge-strand located adjacent to the active site with LptD was crucial not only for proteolysis but also, unexpectedly, for assembly promotion of LptD. Site-directed crosslinking analyses indicated that the unstructured N-terminal half of the β-barrel-forming domain of an immature LptD contacts with the BepA edge-strand. Furthermore, the C-terminal region of the β-barrel-forming domain of the BepA-bound LptD intermediate interacted with a 'seam' strand of BamA, suggesting that BepA recognized LptD assembling on the BAM complex. Our findings provide important insights into the functional mechanism of BepA.

**\*For correspondence:**
yakiyama@infront.kyoto-u.ac.jp

**Present address:** †Division of Biological Science, Nara Institute of Science and Technology, Ikoma, Japan

**Competing interests:** The authors declare that no competing interests exist.

## Introduction

The cell envelope of diderm bacteria is composed of two membranes, namely the inner (cytoplasmic) membrane (IM) and the outer membrane (OM). The intermembrane space, known as periplasmic space, contains a peptidoglycan layer. The OM is the outermost layer of a cell directly facing the external milieu and acts as a selective permeability barrier that prevents the penetration of toxic compounds including antibiotics (*Nikaido, 2003*). The cell surface localization as well as the functional importance of the OM make its components suitable drug targets.

Outer membrane proteins (OMPs), generally exhibiting a β-barrel fold formed by more than eight β-strands, play important roles in maintaining the structural and functional integrity of the OM (*Konovalova et al., 2017*). Therefore, irregularities in OMP biogenesis result in elevated cellular sensitivity to toxic compounds (*Hart et al., 2019*; *Oh et al., 2011*). After synthesis in the cytoplasm and following translocation across the IM to the periplasm through the SecYEG translocon, OMPs are delivered to the OM by periplasmic chaperones such as DegP, Skp, and SurA, and are finally integrated into the OM (*Konovalova et al., 2017*; *Plummer and Fleming, 2016*; *Ricci and Silhavy, 2019*). The OM assembly of OMPs is mediated by the β-barrel assembly machinery (BAM) complex consisting of a β-barrel OMP, BamA, and four lipoproteins, namely BamB, BamC, BamD, and BamE (*Plummer and Fleming, 2016*; *Ricci and Silhavy, 2019*; *Tomasek and Kahne, 2021*). Among the BAM complex subunits, BamA and BamD are the only essential components, although recent studies

have shown that certain BamA mutations render all other BAM subunits dispensable (*Hart et al., 2020*). The BAM complex has a silk-hat-like structure (*Bakelar et al., 2016*; *Gu et al., 2016*; *Han et al., 2016*; *Iadanza et al., 2016*; *Tomasek et al., 2020*); the OM-embedded C-terminal β-barrel domain of BamA forms the 'crown' and the N-terminal periplasmic polypeptide transport-associated (POTRA) domains of BamA form the 'brim' together with the BamB/C/D/E lipoproteins.

Lipopolysaccharide (LPS), another major OM constituent localized in the outer leaflet of the OM, is also important for the maintenance of the structure and function of the OM (*Sperandeo et al., 2017*). LPS is synthesized on the cytoplasmic side of the IM and flipped across the IM to the periplasm. After maturation, it is transported to the OM by the LPS transport (Lpt) proteins (*Sperandeo et al., 2017*). A heterodimer of LptD, a β-barrel OMP, and LptE, a lipoprotein, plays roles in the final step to insert LPS into the OM (*Dong et al., 2014*; *Qiao et al., 2014*; *Wu et al., 2006*). The OM assembly process of LptD is unique in that it is accompanied by the rearrangement of intramolecular disulfide bonds. The mature form of LptD possesses two disulfide bonds formed by non-consecutive pairs of Cys residues (C31–C724 and C173–C725) (*Ruiz et al., 2010*). It has been shown, however, that an assembly intermediate of LptD having disulfide bonds formed by consecutive pairs of the Cys residues (C31–C173 and C724–C725) (LptD$^C$; LptD with **C**onsecutive disulfide bods) is first generated and isomerized to LptD$^{NC}$ (LptD with **N**on-**C**onsecutive disulfide bonds) during its assembly/maturation, which is triggered by the association of LptD with LptE (*Chng et al., 2012*; *Narita et al., 2013*). The LptD$^C$ to LptD$^{NC}$ conversion should occur at a later step in the OM assembly because LptD presumably associates with LptE on the BAM complex (*Chimalakonda et al., 2011*; *Chng et al., 2012*; *Lee et al., 2016*; *Narita et al., 2013*).

BepA (formally called YfgC), a bi-functional periplasmic protein that plays an important role in maintaining OM integrity (*Narita et al., 2013*), belongs to the M48 family zinc-metallopeptidases that include prokaryotic and eukaryotic proteases (such as Ste24, Oma1, and HptX) involved in membrane quality control (*Rawlings et al., 2018*). We have previously shown that BepA is involved in the biogenesis and quality control of LptD. While BepA promotes the LptD$^C$ to LptD$^{NC}$ conversion (chaperone-like function) (*Narita et al., 2013*), it also degrades the stalled or misassembled LptD$^C$ molecules that are generated due to an *lptD* mutation (*lptD4213*) or decreased availability of or weakened interaction with LptE (protease function) (*Narita et al., 2013*; *Soltes et al., 2017*). BepA also degrades BamA whose assembly/folding has been impaired in the absence of a periplasmic chaperone, SurA (*Daimon et al., 2017*), suggesting that BepA can also act in quality control of some other OM proteins. The BepA protein consists of an N-terminal M48 metallopeptidase domain and a C-terminal tetratricopeptide repeat (TPR) domain that are associated closely to form a compact structure (*Figure 1A*; *Bryant et al., 2020*; *Shahrizal et al., 2019*). Our previous study suggested that the TPR domain of BepA directly interacts with LptD and with the BAM complex with its TPR domain inserted into the interior of the periplasmic part (brim) of the BAM complex (*Daimon et al., 2017*; *Narita et al., 2013*). A mutational study has suggested that these interactions are important for BepA functions (*Daimon et al., 2017*). Recent studies have also shown that the His-246 residue of BepA that coordinates the zinc ion at the proteolytic active site acts as an ON/OFF switch (His switch) for the proteolytic activity of BepA (*Bryant et al., 2020*; *Daimon et al., 2020*). The dual functions of BepA should be appropriately regulated because the unregulated expression of the proteolytic activity of BepA caused by an H246 mutation leads to the degradation of the normally assembling LptD intermediate (*Daimon et al., 2020*). However, information on the molecular mechanism of this regulation and the modes of the BepA–LptD interaction in each BepA function remains elusive.

Here, we investigated the mechanism by which BepA established interaction with LptD in the promotion of its assembly and degradation. Our results showed that a conserved β-strand (edge-strand) located adjacent to the BepA active site directly contacts with LptD and plays important roles in substrate proteolysis, like many other proteases. In addition, we unexpectedly found that the edge-strand-mediated interaction with a substrate is also required for the chaperone-like function of BepA, which should be enabled by the His switch-mediated repression of the proteolytic activity. Crosslinking experiments demonstrated that BepA could interact with an LptD molecule assembling on the BAM complex. Based on these observations, we propose a model explaining the edge-strand and His switch-mediated functional regulation of BepA in LptD assembly/degradation.

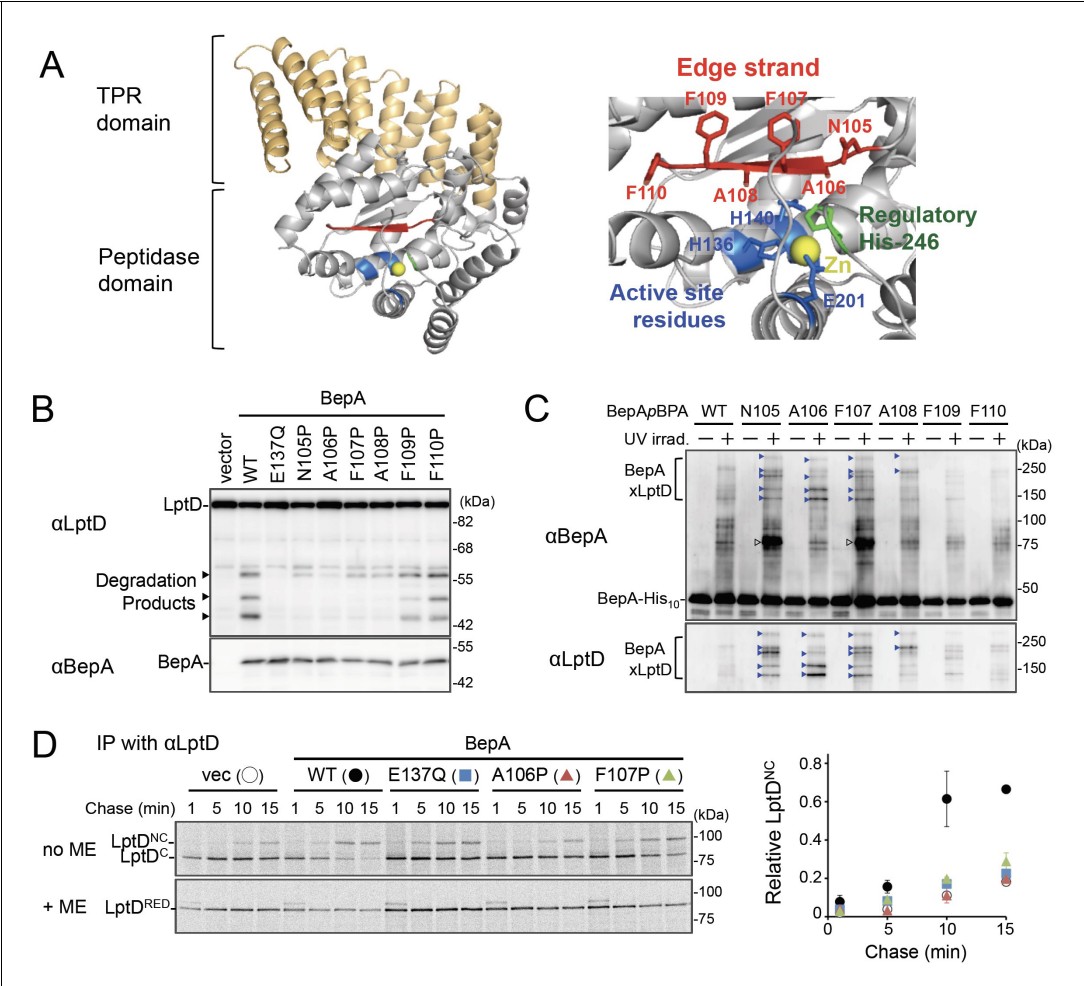

**Figure 1.** The edge-strand of BepA is crucial for functional interaction with LptD. (**A**) Crystal structure of BepA (PDB code: 6AIT). The peptidase and the TPR domains of BepA are shown in gray and orange, respectively. The edge-strand, the proteolytic active site (the HExxH motif and the third zinc ligand, Glu-201), and the regulatory His-246 residue (His switch) in the peptidase domain are shown in red, blue, and green, respectively, and the coordinated zinc atom is shown in yellow. An enlarged view of the active site region is shown in *right*. (**B**) Protease activities of the BepA edge-strand mutants. Cells of SN56 (Δ*bepA*) carrying pTWV228-*lptD-his_{10}* and either pSTD689 or pSTD689-*bepA* plasmids were grown at 30°C in L-medium until early log phase and induced with 1 mM IPTG for 1 hr. Total cellular proteins were acid-precipitated and analyzed by 7.5 or 10% Laemmli SDS-PAGE and immunoblotting with the indicated antibodies. (**C**) In vivo photo-crosslinking analysis of the BepA edge-strand. Cells of SN56 carrying pEVOL-pBpF and pUC18-*bepA(E137Q, amb)-his_{10}* plasmids were grown at 30°C in L-medium containing 0.02% arabinose and 0.5 mM *p*BPA until early log phase, and induced with 1 mM IPTG for 1 hr to express the indicated BepA(*p*BPA) variants. The cultures were divided into two portions, each of which was treated with or without UV-irradiation for 10 min at 4°C. Proteins of the total membrane fractions were subjected to pull-down with Ni-NTA agarose. Purified proteins were analyzed by 7.5% Laemmli SDS-PAGE and immunoblotting with the indicated antibodies. Open triangles indicate unknown crosslinked products. (**D**) Chaperone-like activities of the BepA edge-strand mutants. Cells of SN56 carrying pSTD689 or a pSTD689-*bepA* plasmid were grown at 30°C in M9-based medium until early log phase, induced with 1 mM IPTG for 15 min, pulse-labeled with $^{35}$S-Met for 1 min and chased for the indicated periods. At each time point, total cellular proteins were acid-precipitated, subjected to IP with an anti-LptD antibody, and analyzed by 7.5% Laemmli SDS-PAGE followed by phosphorimaging. The ratio of the band intensities of LptD$^{NC}$ at each time point to that of total LptD (LptD$^{C}$+LptD$^{NC}$) at 5 min was quantitated and the mean values were plotted with S.D. (n=2). The result shown is a representative of two independent experiments that were conducted using the same transformants (i.e., two technical replicates). See *Figure 1—source data 1* for gel images and quantitated band intensities data for (**D**).

The online version of this article includes the following source data and figure supplement(s) for figure 1:

**Source data 1.** A Zip file containing gel images.

**Figure supplement 1.** Sequence alignment of BepA homologs and *Escherichia coli* M48 family peptidases.

**Figure supplement 2.** Effects of the BepA edge-strand Pro mutations on the BamA degradation and the BepA self-cleavage.

**Figure supplement 2—source data 1.** A Zip file containing gel images (**A–C**) for the immunoblotting experiments using the anti-BepA and anti-LptD antibodies.

**Figure supplement 3.** Functionality of the BepA derivatives having *p*BPA in the edge-strand.

*Figure 1 continued on next page*

*Figure 1 continued*

**Figure supplement 3—source data 1.** A Zip file containing plate images (A) and gel images (B) for the immunoblotting experiments using the anti-BepA antibody.

**Figure supplement 4.** The chaperone-like and proteolytic activities of the BepA edge-strand mutants.

**Figure supplement 4—source data 1.** A Zip file containing gel images (A, B) for the immunoblotting experiments using the anti-BepA and anti-BamA antibodies.

## Results

### Interaction of the BepA edge-strand with LptD is crucial not only for proteolysis but also for assembly promotion of LptD by BepA

Zinc-metallopeptidases usually possess a β-strand, called edge-strand, located close to their proteolytic active sites (*Akiyama et al., 2015*; *López-Pelegrín et al., 2013*; *Stöcker and Bode, 1995*). While the edge-strand is known to play a critical role in substrate proteolysis by directly interacting with a substrate polypeptide by the strand addition mechanism and converts it into an extended conformation for its presentation to the active site and proteolysis, it has not been well characterized in M48 proteases. The solved structures of *Escherichia coli* BepA (*Bryant et al., 2020*; *Shahrizal et al., 2019*) show that it has a β-strand (β2) that is conserved among the M48-peptidases and is located adjacent to the active site (*Figure 1A* and *Figure 1—figure supplement 1*), suggesting that this strand presumably acts as an edge-strand. To examine the role of the β2-strand in BepA functions, we constructed BepA mutants by introducing Pro at each position in β2 (from Asn-105 to Phe-110; *Figure 1A*). We then investigated the effects of β2 mutations on the proteolytic activity of BepA against overproduced LptD. When LptD is overproduced from a multi-copy plasmid, it mainly accumulates in the form of $LptD^C$ possibly due to the limited availability of its partner protein, LptE (*Daimon et al., 2020*; *Daimon et al., 2017*). This species probably represents a 'normal' assembly intermediate as it is associated with the BAM complex (see below) and can be converted to the mature form ($LptD^{NC}$) when LptE is co-expressed (*Miyazaki et al., 2018*). As reported previously (*Daimon et al., 2020*; *Daimon et al., 2017*), overproduced LptD was degraded by co-expressed wild-type BepA to generate discrete degradation products (*Figure 1B*). To examine the possible roles of the β2-strand in the function of BepA, we introduced a Pro substitution into the β2-strand, as a Pro residue would affect the secondary structure, and thus the function, of this strand. We found that the expression of a few BepA mutants (N105P, A106P, F107P, and A108P) led to a significantly decreased generation of the LptD degradation products (*Figure 1B*). Furthermore, some of these mutations compromised the degradation of BamA in a Δ*surA* strain (*Daimon et al., 2017*) and the self-cleavage of BepA-$His_{10}$ (BepA possessing a C-terminal $His_{10}$-tag) within the $His_{10}$-tag (*Narita et al., 2013*; *Figure 1—figure supplement 2A and B*). These results strongly suggest that the β2-strand is important for the proteolytic activity of BepA.

We next analyzed whether the β2-strand directly interacts with a substrate by using a site-directed in vivo photo-crosslinking approach (*Chin and Schultz, 2002*; *Miyazaki et al., 2020a*). We expressed derivatives of BepA-$His_{10}$ harboring a photoreactive amino acid analog, *p*-benzoyl-L-phenylalanine (*p*BPA), at each position in the β2-strand in a Δ*bepA* strain and examined their complementation activity regarding erythromycin (EM) sensitivity of the cell and self-cleavage of BepA-$His_{10}$. The results showed that certain mutants (F107*p*BPA, F109*p*BPA, and F110*p*BPA) were as functional as the wild-type, but others exhibited neither significant complementation activity nor self-cleavage (*Figure 1—figure supplement 3A and B*). For the photo-crosslinking experiments conducted in this study, we used BepA variants harboring the E137Q mutation in the $H^{136}ExxH$ motif for two reasons. First, this mutation would repress the possible degradation of LptD by the BepA derivatives. Second, our previous study has shown that this mutation stabilizes the interaction of BepA with LptD (*Daimon et al., 2017*). Following UV-irradiation of cells expressing each of the BepA mutants, BepA-$His_{10}$ and its crosslinked products were purified from the membrane fractions by affinity isolation using Ni-NTA agarose and subjected to immunoblotting analysis (*Figure 1C*). The N105*p*BPA, A106*p*BPA, F107*p*BPA, and A108*p*BPA derivatives of BepA generated, in a UV-dependent manner, evident crosslinked products that reacted with both anti-BepA and anti-LptD antibodies. Taken together, these results imply that the β2-strand of BepA directly interacts with LptD and plays an important role in their degradation. The structure, intramolecular disposition, as

well as involvement in substrate interaction and proteolysis of the β2-strand strongly supports the hypothesis that it indeed acts as the edge-strand to recognize and present a substrate to the active site. We have thus referred to the β2-strand as the 'edge-strand' hereafter. BepA N105*p*BPA and F107*p*BPA derivatives also generated the unidentified crosslinked products (*Figure 1C*, open arrowheads), suggesting that the edge-strand would interact with some other substrates. The differential effects of the mutations on the degradation of LptD and BamA and the self-cleavage of the C-terminal tag (*Figure 1B* and *Figure 1—figure supplement 2A and B*) might reflect their different interaction properties and/or affinities for the BepA edge-strand. Our previous study also suggested substrate-specific interaction of an edge-strand for another metallopeptidase, RseP (*Akiyama et al., 2015*).

BepA not only degrades the stalled or misassembled LptD (when LptD is on the off-pathway) but also promotes the maturation of LptD through facilitating its assembly with the partner protein LptE (chaperone-like activity) (on the normal (on-) pathway) (*Narita et al., 2013*). Therefore, we examined the effects of the edge-strand Pro mutations on the chaperone-like activity of BepA. LptD possesses two intramolecular disulfide bonds. While the mature (fully assembled) LptD has non-consecutive disulfide bonds, an assembly intermediate form of LptD (LptD$^C$) that accumulates in the absence of functional BepA possesses consecutive disulfide bonds. Consistent with our previous results (*Narita et al., 2013*), the expression of the wild-type BepA in a Δ*bepA* strain markedly decreased the accumulation of LptD$^C$ (*Figure 1—figure supplement 4A*). On the contrary, only a partial decrease in accumulation was observed with the expression of the E137Q mutant. We found that the A106P mutant was defective in reducing LptD$^C$ accumulation, suggesting that a mutation in the edge-strand could affect the chaperone-like function (*Figure 1—figure supplement 4A*). Subsequently, we examined the effects of two edge-strand mutations—A106P and F107P—that markedly compromised LptD degradation on the chaperone-like function of BepA using pulse-chase experiments. Pulse-chase analysis showed that LptD was largely stable, but received a slight degradation during the chase period under the condition used (*Figure 1D*, left panel, +ME). It is likely that some of LptD$^C$ was degraded by BepA during the chase period, because LptD$^{NC}$ is largely BepA-resistant (*Daimon et al., 2020*). Thus, the ratio of LptD$^{NC}$ to the total LptD (LptD$^C$+LptD$^{NC}$) at each chase point does not exactly reflect the efficiency of the LptD$^C$-to-LptD$^{NC}$ conversion of pulse-labeled LptD. To examine only the chaperone-like activities (promotion of the LptD$^{NC}$ generation) without the possible effect of the LptD degradation by the residual proteolytic activities of the A106P and the F107P, the relative amounts of LptD$^{NC}$ at each chase point to the total LptD (LptD$^C$+LptD$^{NC}$) at an early chase point (5 min) were calculated and plotted as quantified data (*Figure 1D*, right panel). The expression of the wild-type BepA in Δ*bepA* cells significantly accelerated the conversion of LptD$^C$ to LptD$^{NC}$, whereas the expression of the protease-dead E137Q mutant demonstrated weakened conversion (*Figure 1D*). The acceleration of the LptD$^C$ to LptD$^{NC}$ conversion by the F107P and the A106P mutants was much weaker than that by the wild-type BepA (almost the same as compared with the E137Q mutant) (*Figure 1D*). These results unexpectedly suggested that the edge-strand was also important for the chaperone-like activity of BepA.

We constructed BepA derivatives with a Cys substitution (note that BepA intrinsically possesses no Cys residue) at the position of Asn-105, Ala-106, or Phe-107 in the edge-strand to examine whether a specific residue in this strand is required for the BepA's functions. We chose a Cys substitution, because (i) the Cys mutants can be used in the disulfide crosslinking experiments described below, and (ii) a previous study strongly suggested that a Cys mutation does not affect the secondary structure of an edge-strand in another protease RseP (*Akiyama et al., 2015*). These BepA Cys constructs exhibited almost normal chaperone-like and proteolytic functions (*Figure 1—figure supplement 4B and C*). This supports the idea that the secondary structure of the edge-strand is more important than the individual amino acid residues for its function, although some contribution of the side-chains of the amino acids to the functions of the edge-strand cannot be excluded. Note that, although it is possible that a Pro mutation also affects the structures of the overall BepA protein and/or the active site around the edge-strand, the edge-strand Pro mutants (other than F107P) still exhibited significant self-cleavage of the probably unstructured-terminal tag (*Figure 1—figure supplement 2B*). In addition, the Pro mutants (other than A106P) degraded mis- or un-folded BamA at a detectable level (*Figure 1—figure supplement 2B*). Together with the result that these mutants accumulated at a level comparable to that of wild-type BepA, the above observations suggest that

most of the Pro mutations specifically affected the edge-strand structure, but not drastically altered the active site or the protein's overall structures.

## BepA interacts with the N-terminal half of the β-barrel-forming domain of the LptD assembly intermediate

While BepA interacts with LptD to promote either its proper OM assembly or proteolytic elimination depending on the situation (*Narita et al., 2013*), the details of the BepA–LptD interaction, including the region(s) in LptD to which BepA binds, remain largely unknown. Thus, we performed a systematic photo-crosslinking analysis to identify the BepA-contact region in the LptD assembly intermediate LptD$^C$. We performed photo-crosslinking experiments in cells ectopically co-expressing an LptD derivative containing a photo-reactive amino acid analog pBPA [LptD(pBPA)] and a protease-dead variant of BepA, BepA(E137Q). We first introduced pBPA at each of the 50 positions (approximately every 15 residues) in the mature part of LptD and performed photo-crosslinking analysis. Cells expressing LptD(pBPA)-His$_{10}$ and BepA(E137Q) were grown and UV-irradiated, and the total cellular proteins were analyzed by immunoblotting with anti-BepA and anti-His antibodies. Under this condition, expressed LptD-His$_{10}$ was considerably accumulated as LptD$^C$ (*Figure 2—figure supplement 1*) irrespective of co-expression of BepA(E137Q). We detected clear crosslinking with BepA mainly in the N-terminal half of the LptD β-barrel-forming domain (*Figure 2A*). We then performed a detailed photo-crosslinking analysis for the 20 additional sites in the N-terminal half of the LptD β-barrel-forming domain (*Figure 2B*) and found that BepA was crosslinked at several of these sites. The residues at the BepA-cross-linkable sites were oriented both inward and outward in the mature LptD β-barrel (*Figure 2C and D*). Moreover, the residue Gln-393 at which the strongest crosslinking was observed was oriented inward. These results suggest that LptD while interacting with BepA would not assume a higher-order structure like closed β-barrel (see Discussion). We selected a few LptD (pBPA) derivatives that had been crosslinked with BepA as representatives and examined their functionality. They supported the growth of LptD-depleted cells when expressed from a plasmid, indicating that they were functional (*Figure 2—figure supplement 2*). The above-mentioned crosslinking results thus likely reflect a functional interaction of LptD with BepA in the normal assembly pathway.

## BepA edge-strand directly interacts with the Tyr-331 residue in the β7 strand of the LptD β-barrel domain

We investigated further to identify the region of LptD that interacts with the BepA edge-strand. First, we examined the effects of the BepA edge-strand Pro mutations (F107P and A106P) on the LptD(pBPA)–BepA crosslinking. The F107P mutation significantly decreased the efficiency of the crosslinking of BepA(N105pBPA) and BepA(A106pBPA) with LptD (*Figure 3—figure supplement 1*). Additionally, the A106P mutation exhibited a similar effect on the crosslinking of BepA(N105pBPA) with LptD. Based on these effects on crosslinking, we inferred that these mutations affected the interaction of the edge-strand with LptD. Subsequently, we selected several LptD(pBPA) derivatives that showed relatively strong crosslinking with BepA and examined the effect of F107P and A106P mutations in the BepA edge-strand on the crosslinking of LptD(pBPA) with BepA. We found that these mutations altered LptD–BepA crosslinking in a site-specific manner. Further, the amount of crosslinked products markedly decreased for LptD(Y331pBPA), but not for other mutants (*Figure 3A*). These results strongly suggest that the region around Tyr-331 in the β7 strand of the LptD β-barrel domain (*Figure 2D*) is crosslinked with the edge-strand of BepA. Note that, while we detected LptD-BepA crosslinked products ranging from 150 to 250 kDa with BepA derivatives having pBPA in the edge-strand (*Figure 1C*), we did not observe similar multiple crosslinked products with LptD(Y331pBPA) (*Figure 2*). The exact reason for this is unknown, but it might be ascribed to the crosslinking of pBPA in the BepA edge-strand to different LptD positions that are close spatially but distant in the primary sequence, which could generate crosslinked products with different mobility (i.e., different apparent sizes).

To further confirm the direct interaction of the BepA edge-strand with the LptD β7 strand, we conducted site-specific disulfide crosslinking experiments. For this analysis, we used the above-described single Cys derivatives of BepA harboring a Cys residue at the position of Asn-105, Ala-106, or Phe-107, and derivatives of LptD having a Cys substitution at either of the six positions, including Tyr-331 at which introduction of pBPA showed clear crosslinking with BepA (*Figure 3A*).

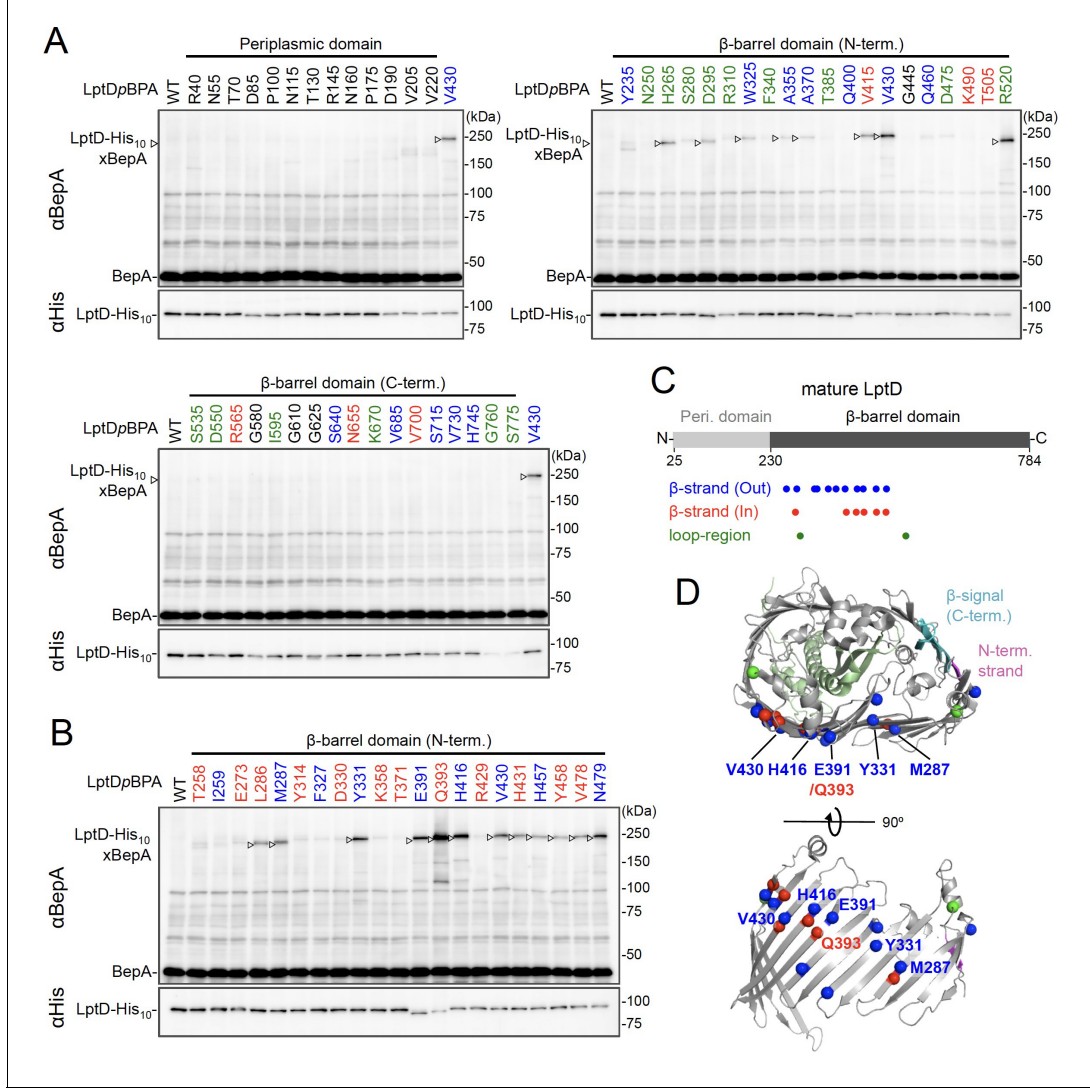

**Figure 2.** Photo-crosslinking of the β-barrel forming domain of LptD with BepA. (**A, B**) In vivo photo-crosslinking between LptD and BepA. Cells of RM2243 (*bepA(E137Q)*) carrying pEVOL-pBpF, pMW118-*bepA(E137Q)*, and pRM294-*lptD(amb)-his10* plasmids were grown at 30˚C in L-medium containing 0.02% and 0.5 mM *p*BPA until early log phase and induced with 1 mM IPTG for 3 hr to express the indicated LptD(*p*BPA) variants. The cultures were then divided into two portions, each of which was UV-irradiated for 10 min at 4˚C. Total cellular proteins were acid-precipitated and analyzed by 7.5% Laemmli SDS-PAGE and immunoblotting with the indicated antibodies. Most of the LptD mutants were accumulated in comparable amounts. LptD-His10xBepA crosslinked products were not detectable with an anti-His antibody due to its low reactivity to LptD-His10 in this and the following experiments. Amino acid residues shown in red and blue indicate the ones whose side chain is pointing inward and outward, respectively. Amino acid residues shown in green indicate the ones located in the loop regions. The result shown is a representative of two technical replicates. (**C**) Summary of the BepA crosslinked positions in LptD. Positions where the crosslinking with BepA was clearly and reproducibly detected are indicated by colored dots. (**D**) Mapping of the BepA crosslinked positions on the barrel domain of LptD in the *Escherichia coli* LptD–LptE structure (PDB code: 4RHB). LptD and LptE are shown in gray and light green, respectively. The N-terminal strand and the β-signal (C-terminal region) in the LptD β-barrel domain are shown in magenta and light blue, respectively. The top view of the LptD/E structure from extracellular space (*upper*) and the side view of the N-terminal region of LptD β-domain (*lower*) are shown. The positions where the crosslinking with BepA was observed were indicated by spheres colored as above. See *Figure 2—source data 1* for gel images for (**A, B**).

The online version of this article includes the following source data and figure supplement(s) for figure 2:

**Source data 1.** A Zip file containing gel images (A, B) for the immunoblotting experiments using the anti-BepA and anti-His tag antibodies.

**Figure supplement 1.** Overexpressed LptD molecules mainly accumulate as LptD^C.

**Figure supplement 1—source data 1.** A Zip file containing gel images for the immunoblotting experiments using the anti-LptD, anti-BepA, and anti-BamA antibodies.

**Figure supplement 2.** Complementation activity of the LptD(*p*BPA) derivatives.

**Figure supplement 2—source data 1.** A Zip file containing plate images for the complementation assays for the LptD(*p*BPA) derivatives.

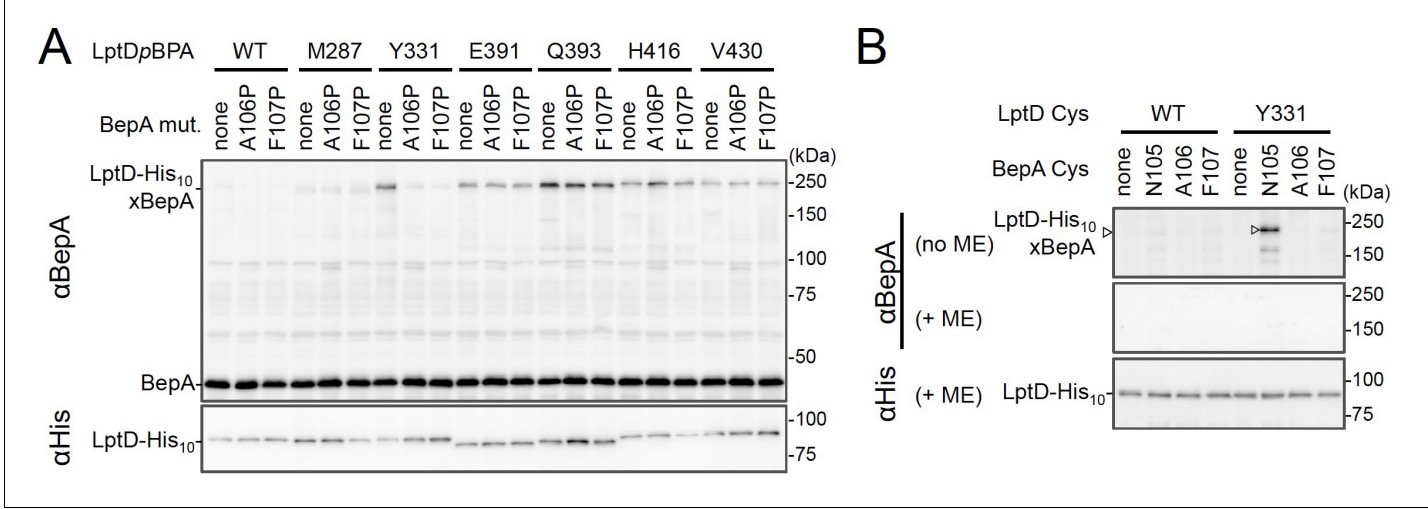

**Figure 3.** BepA edge-strand directly contacts with the Tyr-331 residue in the N-terminal half region of the LptD β-barrel forming domain. (**A**) Effect of the BepA edge-strand mutations on the crosslinking between BepA and the LptD derivatives having pBPA in the N-terminal half region of the LptD β-barrel-forming domain. Cells of SN56 (ΔbepA) carrying pEVOL-pBpF, pMW118-bepA(E137Q, mut), and pRM294-lptD(amb)-his$_{10}$ were grown, induced to express a BepA and a LptDpBPA derivative, and subjected to photo-crosslinking analysis as described in *Figure 2*. (**B**) Disulfide crosslinking between the Cys residues in the edge-strand of BepA and the N-terminal half region of the LptD β-barrel-forming domain. Cells of SN56 (ΔbepA) carrying a combination of plasmids encoding WT or a Cys-introduced mutant of BepA and LptD-His$_{10}$ as indicated were grown in L-medium and induced with 1 mM IPTG for 3 hr to express BepA(Cys) and LptD(Cys)-His$_{10}$. Total cellular proteins were acid-precipitated, solubilized with SDS buffer containing NEM (for blocking free thiol groups), and subjected to pull-down with Ni-NTA agarose. The purified proteins were treated with or without 2-mercaptoethanol (ME) and analyzed by 7.5% Laemmli SDS-PAGE and immunoblotting with the indicated antibodies. The result shown is a representative of two technical replicates. See *Figure 3—source data 1* for gel images for (**A, B**).

The online version of this article includes the following source data and figure supplement(s) for figure 3:

**Source data 1.** A Zip file containing gel images (**A, B**) for the immunoblotting experiments using the anti-BepA and anti-His tag antibodies.

**Figure supplement 1.** Effects of the BepA edge-strand mutations on the crosslinking of the BepA edge-strand to LptD.

**Figure supplement 1—source data 1.** A Zip file containing gel images for the immunoblotting experiments using the anti-LptD, anti-BepA, and anti-BamA antibodies.

**Figure supplement 2.** Disulfide crosslinking between BepA and LptD.

**Figure supplement 2—source data 1.** A Zip file containing plate images (**A**) for the complementation assays for the LptD(Cys) derivatives and gel images (**B**) for the immunoblotting experiments using the anti-BepA and anti-His antibodies.

The wild-type LptD protein harbors intrinsic four Cys residues that form two disulfide bonds essential for the LptD function; therefore, each of these Cys-substituted LptD mutants possessed five Cys residues in total. We confirmed that these Cys-substituted LptD derivatives accumulated normally and retained their function (*Figure 3—figure supplement 2*). Cells expressing a combination of BepA Cys mutants and LptD Cys mutants were grown, and total proteins were acid-denatured and dissolved in SDS containing N-ethylmaleimide (NEM; NEM was included to block free Cys residues). Then, LptD-His$_{10}$ and its crosslinked products were affinity-isolated using the C-terminal His$_{10}$-tag, treated with or without 2-mercaptoethanol (ME), and analyzed by SDS-PAGE and anti-BepA immunoblotting. We observed that certain combinations of BepA and LptD derivatives showed a high-molecular-mass band in electrophoresis results. Among them, the combination of BepA(N105C) and LptD(Y331C) showed the most intense band that exhibited reaction with the anti-BepA antibody (*Figure 3B* and *Figure 3—figure supplement 2B*, no ME). These high-molecular-mass bands were not observed with the wild-type LptD (no additional Cys) and disappeared upon treatment with ME, suggesting that they were disulfide-crosslinked products (*Figure 3B* and *Figure 3—figure supplement 2B*, + ME). These results are consistent with the photo-crosslinking experiments (*Figure 3A*) and indicate that the edge-strand of BepA can directly bind to several regions in the N-terminal half of the LptD β-barrel-forming domain, which includes the β7 strand containing Tyr-331.

## BepA interacts with an LptD intermediate associating with the seam region of BamA on the BAM complex

We further investigated the mode of the interaction of the BepA-associated LptD with the BAM complex. It has been recently shown that LptD4213, a mutant form of LptD that has a short (23 amino acids) deletion in an extracellular loop (eL4) and is stalled on the BAM complex mimicking a late assembly complex (*Lee et al., 2016*), interacts with the seam region formed by the N- and C-terminal β-strands (β1 and β16, respectively) in the BamA barrel domain and forms a hetero-complex, in which the C-terminal β-signal of the LptD4213 was associated with the β1 strand of the BamA seam (*Lee et al., 2019*). It has been suggested that the interaction of the β-signal with the BamA β1 strand generally facilitates the folding of the β-barrel domain of a substrate OMP (see *Figure 4D* and *5B*; *Tomasek and Kahne, 2021*). We first examined the interaction of the BepA-associated LptD intermediate with BamA and BamD by conducting in vivo photo-crosslinking experiments using the LptD derivatives with *p*BPA at the position of Glu-749 or Tyr-726 in addition to the position of Tyr-331. *p*BPA at Glu-749 and Tyr-726 residue, both of which are located near the β-signal of LptD, have been reported to be crosslinked with BamA and BamD, respectively, during the LptD assembly (*Figure 4D*; *Lee et al., 2019*; *Lee et al., 2018*). Complementation assay results showed that either of the LptD derivatives containing one or two *p*BPA at Glu-749 and Tyr-726 were functional (*Figure 4—figure supplement 1*). After UV-irradiation of the cells expressing LptD-His$_{10}$ and the *p*BPA derivatives, LptD-His$_{10}$ and its crosslinked products were affinity-isolated from the membrane fractions and analyzed by immunoblotting. The single *p*BPA derivatives indeed generated crosslinked products with the expected factors as: LptD(Y331*p*BPA) with BepA, LptD(D749*p*BPA) with BamA, and LptD(Y726*p*BPA) with BamD (*Figure 4A and B*). With the double *p*BPA derivatives LptD(Y331/D749*p*BPA) and LptD(Y331/Y726*p*BPA), new crosslinked products with higher molecular sizes were observed in addition to the ones observed with single *p*BPA mutants (*Figure 4A and B*). These results showed that the higher molecular-sized products represented BepA–LptD(Y331/D749*p*BPA)–BamA and BepA–LptD(Y331/Y726*p*BPA)–BamD crosslinked products and that BepA interacts with an assembly intermediate of LptD on the BAM complex.

We further conducted photo-crosslinking experiments using LptD locked on the BAM complex by using an SH-crosslinker, 1,4-bismaleimidobutane (BMB). It has been previously shown (*Lee et al., 2019*) that a cysteine placed near the N-terminal β-strand of the BamA β-barrel (S439C) was crosslinked with a cysteine introduced near the β-signal of LptD (E733C) via BMB treatment (*Figure 4D*; *Figure 4—figure supplement 2A*). We introduced a Y331*p*BPA mutation into LptD(E733C) and confirmed that the resultant mutant was functional (*Figure 4—figure supplement 2B*). When the LptD(Y331*p*BPA/E733C) mutant was expressed in a strain having a chromosomal *bamA(S439C)* mutant gene, LptD–BepA crosslinked products were detected upon UV-irradiation, whereas LptD–BamA crosslinked products was detected upon BMB treatment. When cells were first treated with BMB and then UV-irradiated, a higher mass product that reacted with both anti-BepA and anti-BamA antibodies was generated. The generation of this product depended on both BMB-treatment and UV-irradiation (*Figure 4C*). These results are fully consistent with the above photo-crosslinking results and further demonstrated that BepA could interact with an LptD assembly intermediate associating with the seam site of BamA on the BAM complex.

## Discussion

The involvement of BepA in the maintenance of structural and functional integrity of the OM was first suggested on the basis of increased sensitivities of the Δ*bepA*(*yfgC*) strain to several antibiotics and chemicals, which was similar to the characteristics of strains with a disruption of genes encoding proteins engaged in outer membrane biogenesis (*Narita et al., 2013*; *Ruiz et al., 2005*; *Tamae et al., 2008*). We have previously shown that BepA is involved in the biogenesis and quality control of LptD, probably on the BAM complex (*Daimon et al., 2020*; *Daimon et al., 2017*; *Narita et al., 2013*). However, the mechanism by which BepA recognizes and interacts with LptD remains elusive. This information is important to understand the mechanism of BepA to distinguish between the normal (on-) and off-pathway intermediates of LptD that are either assembled into the OM or degraded, respectively.

To gain insight into the BepA–LptD interaction, we examined the role of the conserved edge-strand of BepA in its function. Our results showed that the BepA edge-strand participates not only

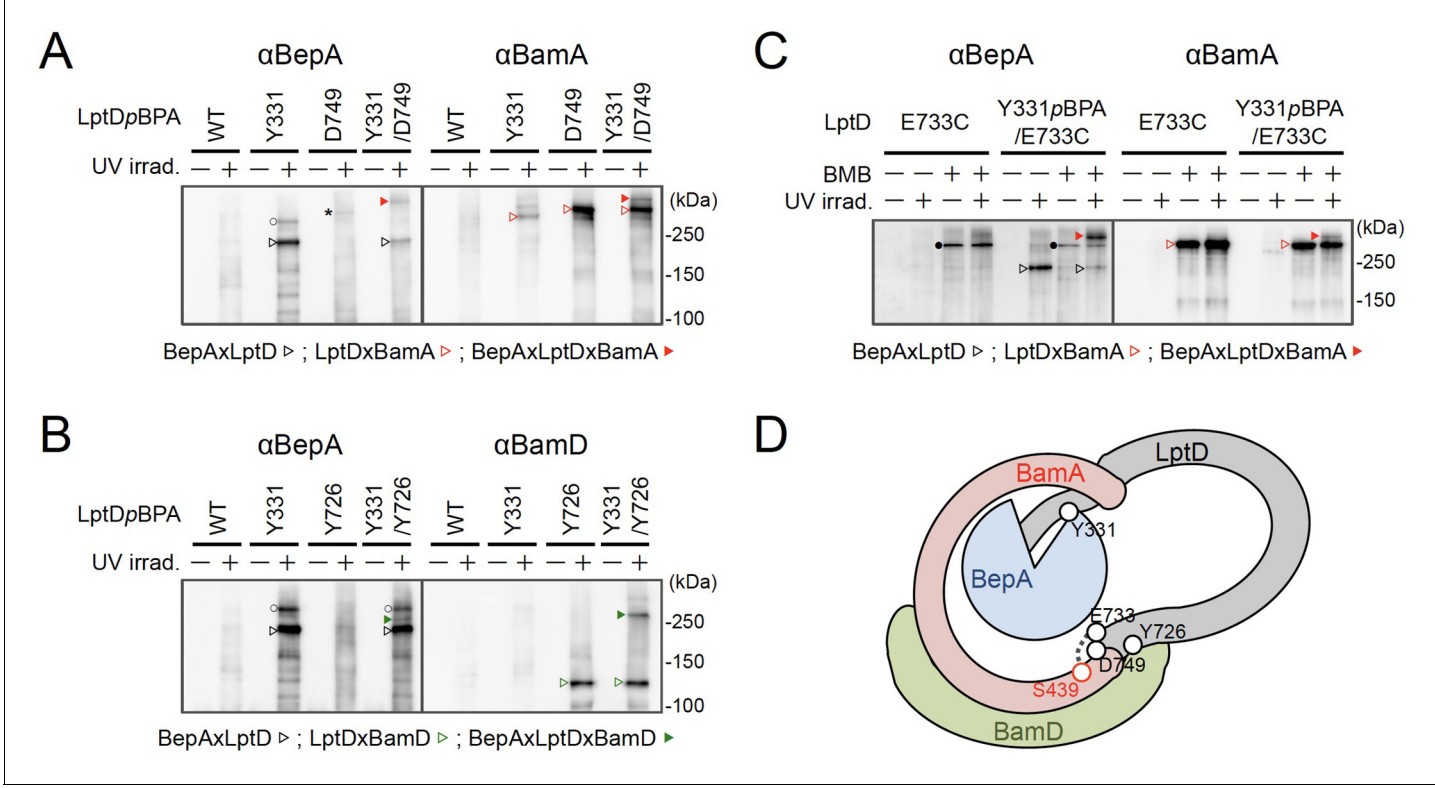

**Figure 4.** BepA interacts with an LptD intermediate assembling on the BAM complex. (**A, B**) In vivo photo-crosslinking of an LptD mutant having *p*BPA at two positions with BepA. Cells of RM2243 (*bepA(E137Q)*) carrying pEVOL-pBpF, pMW118-*bepA(E137Q)* and pRM294-*lptD(amb)-his*$_{10}$ plasmids were grown at 30°C in L-medium containing 0.5 mM *p*BPA until early log phase and induced with 1 mM IPTG for 3 hr to express the indicated LptD(*p*BPA) variants. The cultures were divided into two portions, each of which was treated with or without UV-irradiation for 30 min at 4°C. Proteins of the total membrane fractions were subjected to pull-down with Ni-NTA agarose. Purified proteins were analyzed by 7.5% Laemmli SDS-PAGE by immunoblotting with the indicated antibodies. Asterisk in the anti-BepA blots possibly indicates an LptD-BamA crosslinked product that was detected due to the apparent cross-reactivity of the anti-BepA antibody with the LptDxBamA crosslinked product (see below). (**C**) Simultaneous crosslinking of LptD having Y331*p*BPA and E733C with the BepA edge-strand and the seam region of BamA(S439C). Cells of RM3655 (*bamA(S439C)*, Δ*bepA*)/pEVOL-pBpF/pMW118-*bepA(E137Q)* carrying pRM294-*lptD(E733C)-his*$_{10}$, or pRM294-*lptD(Y331amb, E733C)-his*$_{10}$ were grown and induced as in (**A**). After treatment with or without BMB and the following quenching of BMB by addition of excess cysteine, the cultures were divided into two portions, each of which was treated with or without UV-irradiation for 30 min at 4°C. Total cellular proteins were acid-precipitated, solubilized with SDS buffer containing NEM, and subjected to pull-down with Ni-NTA agarose. Purified proteins were analyzed by 7.5% Laemmli SDS-PAGE and immunoblotting with the indicated antibodies. The anti-BamA immunoblotting showed that the amount of the BepAxLptDxBamA crosslinked product was much lower than that of the LptDxBamA crosslinked product. Although the anti-BepA antibodies apparently cross-reacted weakly with the LptDxBamA crosslinked products (closed circles), the higher signal intensity of the BepAxLptDxBamA crosslinked product band as compared with the intensity of the LptDxBamA band (closed circles) indicate that the detection of the former band with the anti-BepA antibodies cannot be ascribed to this cross-reactivity. The identities of the bands marked by open circles in (**A, B**) are unclear; they might represent BepA-LptD crosslinked products or BepA-BamA crosslinked products (detected due to the cross-reactivity of anti-BepA antibodies with LptDxBamA crosslinked products as described above). In (**A–C**), we confirmed that the amounts of the isolated non-crosslinked LptD-His$_{10}$ derivatives were roughly equal by CBB staining or anti-His immunoblotting (*Figure 4—figure supplement 3*). The result shown is a representative of two technical replicates. (**D**) A schematic cartoon of the interaction of the LptD assembly intermediate with BepA and BamA/D on the BAM complex. See *Figure 4—source data 1* for gel images for (**A–C**).

The online version of this article includes the following source data and figure supplement(s) for figure 4:

Source data 1. A Zip file containing gel images (**A–C**) for the immunoblotting experiments using the anti-BepA, anti-BamA, and anti-BamD antibodies.

Figure supplement 1. Complementation activity of the LptD derivatives having *p*BPA at one or two positions.

Figure supplement 1—source data 1. A Zip file containing plate images for the complementation assays for the LptD(*p*BPA) derivatives.

Figure supplement 2. BMB crosslinking between LptD(Y331*p*BPA/E733C) and BamA(S439C).

Figure supplement 2—source data 1. A Zip file containing gel images (**A**) for the immunoblotting experiments using the anti-BamA and anti-His antibodies and plate images (**B**) for the complementation assays for the LptD(*p*BPA/Cys) derivatives.

Figure supplement 3. The amount of affinity purified LptD-His$_{10}$ derivatives.

Figure supplement 3—source data 1. A Zip file containing stained gel images.

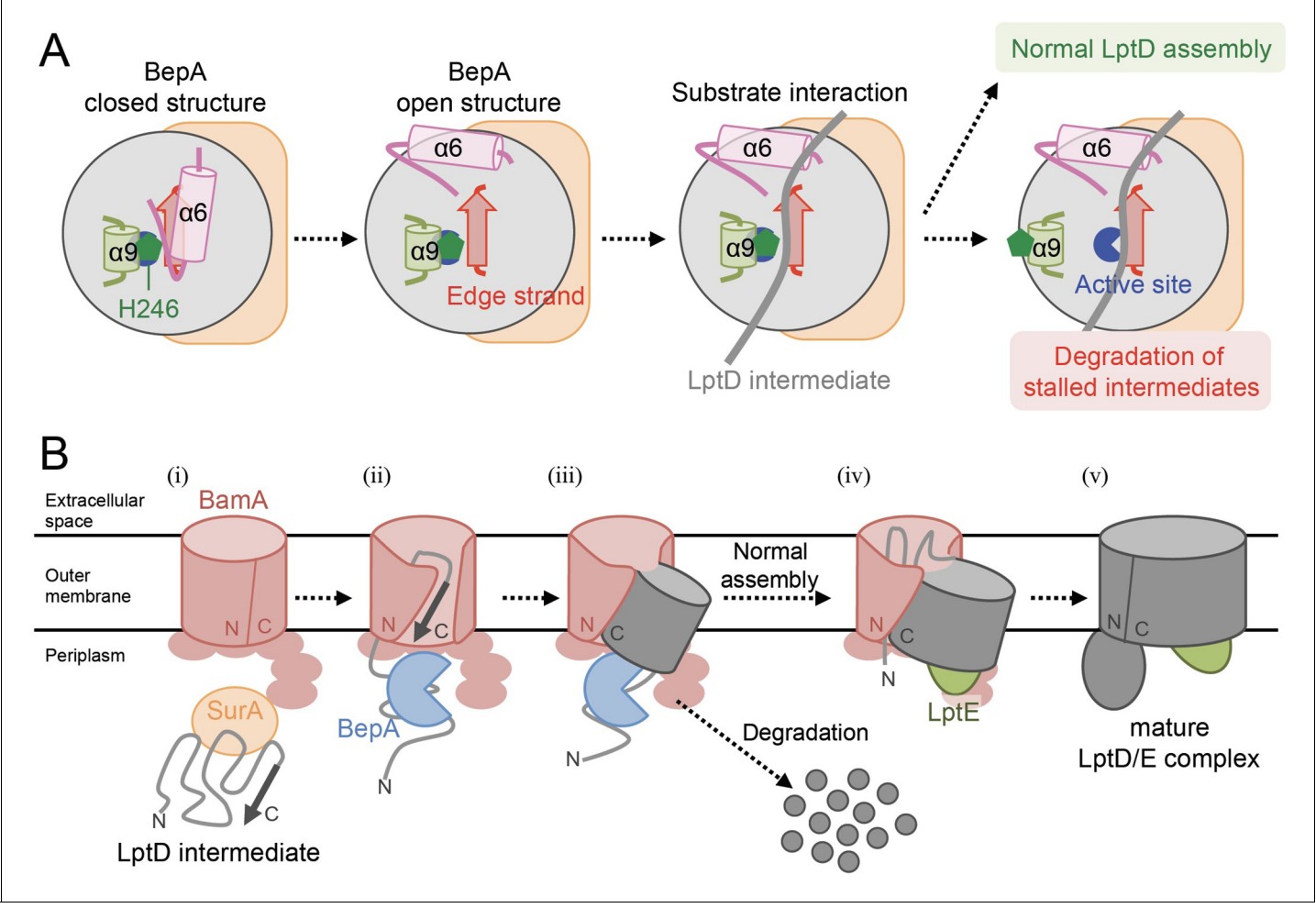

**Figure 5.** Model for the substrate recognition and discrimination by BepA. (**A**) A schematic cartoon of the substrate recognition by BepA at its active site. See the text for details. (**B**) An overview of the proposed LptD assembly process and BepA-mediated discrimination of the assembling and stalled LptD species. See the text for details. Association of BepA with the assembly intermediate form of LptD on the BAM complex could transiently stabilize the LptD assembly intermediate and facilitate the association of LptE with LptD.

The online version of this article includes the following figure supplement(s) for figure 5:

**Figure supplement 1.** α6- and α9-loop regions shield the intramolecular active site and edge-strand of BepA.

in the proteolytic activity but also, unexpectedly, in the chaperone-like activity through its direct interaction with LptD (*Figure 1*). The results of the photo- and disulfide-crosslinking analyses indicated that the N-terminal half of the LptD β-barrel-forming domain interacts with the BepA edge-strand (*Figure 2* and *Figure 3*). Moreover, we showed that BepA demonstrated interaction with an LptD assembly intermediate whose C-terminal region was associated with the seam strand of BamA on the BAM complex (*Figure 4*). A ternary complex formation among an assembly intermediate of an OM protein (EspP, an autotransporter), a periplasmic chaperone (SurA/Skp), and the BAM complex has been suggested form biochemical studies including crosslinking, although the ternary complex was not directly detected (*Ieva et al., 2011*). We here experimentally demonstrated the ternary complex formation for LptD, BepA, and the BAM complex. A similar client-chaperone-BAM ternary complex might be formed in the assembly of other OM proteins. These observations provide useful insights into the BepA functions involved in the biogenesis and quality control of LptD, and also the assembly mechanism of other OM proteins.

## Interaction between the edge-strand of BepA and an LptD assembly intermediate

Our previous results have shown that BepA interacts with the BAM complex via its C-terminal TPR domain partly inserted into the periplasmic ring-like structure of the BAM complex (*Daimon et al., 2017*; *Shahrizal et al., 2019*). In this study, we observed that BepA showed crosslinking with the N-terminal half of the LptD β-barrel-forming domain but not with the C-terminal half. This observation was consistent with the localization of LptD on the BAM complex; the N-terminal region was localized at/near the periplasmic surface of the BAM complex, and the C-terminal region was inserted deep into the BAM complex's interior. The results of the disulfide-crosslinking experiments showed that several positions, including Tyr-331, in the N-terminal half of the LptD β-barrel-forming domain interacted with the edge-strand. The recently solved structures of BepA (*Bryant et al., 2020*; *Shahrizal et al., 2019*) showed that the active site region including the edge-strand was located inside the BepA molecule, leading to the suggestion that structural changes in BepA including the movement of the α6- and α9-loops covering the active site/edge-strand were necessary to enable access of a substrate to the active site/edge-strand region (*Figure 5—figure supplement 1*). However, even after such structural changes, the edge-strand should be located at a recessed position. This suggests that the regions around the BepA-crosslinked positions in the LptD assembly intermediate do not form an extensive β-sheet structure as found in the mature LptD to gain access to the edge-strand of BepA (*Figure 5—figure supplement 1*). Furthermore, *p*BPA at positions of both the inward-pointing and outward-pointing residues in the mature β-barrel domain of LptD was crosslinked with BepA.

Collectively, these results support the hypothesis that the BepA-interacting region of the LptD intermediate is largely unstructured. The unstructured nature of these proteins is in fact helpful to accommodate in or around the narrow space inside the BAM complex. The TPR domain of BepA has also been shown to contact with LptD to promote its biogenesis and degradation (*Daimon et al., 2017*). Currently, we have no information available on the part of LptD that interacts with the TPR domain. The TPR domain may act either together with the edge-strand at the same step or independently at other steps during LptD assembly and degradation.

## The edge-strand and His switch mediate functional regulation of BepA in the assembly promotion and proteolytic quality control of LptD

Further, our results suggest that the proper interaction of LptD with the edge-strand of BepA is important for the promotion of its assembly as well as degradation by BepA (*Figure 1* and *Figure 5A*). This finding was unexpected as it raised a question of how the degradation of the normally assembling LptD intermediate can be avoided despite its interaction with the edge-strand near the protease active site. In the BepA structures, the conserved His-246 residue is coordinated to a zinc ion in the active site to block the activation of a water molecule necessary for the catalysis of the proteolytic reaction (*Bryant et al., 2020*; *Shahrizal et al., 2019*). We have recently reported that His-246 acts as a switch to regulate the proteolytic activity of BepA (*Daimon et al., 2020*). This His-switch-mediated repression of the protease activity would enable the interaction of LptD intermediates at the protease active site of BepA without degradation. Indeed, the derepressed BepA (H246A) mutant degrades an LptD intermediate on the normal assembly pathway, instead of promoting its biogenesis (*Daimon et al., 2020*).

While the exact role of the BepA's edge-strand in the chaperone-like activity remains unclear, our results raise the possibility that the edge-strand-mediated binding of a substrate is involved in the discrimination of LptD intermediates by BepA for assembly or degradation. One possible model would be as follows (*Figure 5A*). In both assembly and degradation pathways of LptD, an unknown signal(s), such as a specific interaction of BepA with the substrate and/or the BAM complex, induces a structural change in BepA (dislocation of the α6-loop covering the active site) to expose its active site and to enable the interaction between the LptD polypeptide and the edge-strand, whereas His-246 continues to repress the degradation of LptD by BepA. This interaction is necessary for both proper assembly and degradation of LptD. The transient interaction of normally assembling LptD with the 'protease activity-repressed' state of BepA would provide sufficient time for the progression of the maturation processes of LptD (including association with LptE), resulting in the final release of mature LptD from BamA. In contrast, during the prolonged stay of stalled or misassembled LptD

molecules, BepA undergoes a further structural change including the movement of the α9-loop, stochastically or induced by a specific signal(s) that was transmitted from the BAM complex and/or LptD through their interaction with BepA, to remove His-246 from the active site zinc, leading to the degradation of the molecules that cannot be subjected to maturation successfully. Our previous results of pulse-chase experiments showed that, although the conversion of LptD$^C$ to LptD$^{NC}$ was relatively slow, it started immediately after the synthesis of LptD and proceeded rather constantly (*Narita et al., 2013*). In contrast, degradation of stalled LptD in an LptE-depleted strain apparently occurred in a biphasic manner; little degradation of stalled LptD was observed for several minutes after synthesis (*Narita et al., 2013*), indicating the presence of a lag period for degradation of stalled LptD to occur. These observations are consistent with the above model.

Notably, it has been reported that BepA expression can result in the degradation of a mutant form of LptD (LptD4213) that probably mimics a late-state assembly intermediate of LptD (*Lee et al., 2019*; *Lee et al., 2016*) that possesses a substantial degree of a higher order (β-barrel-like) structure and interacts with the Bam components (BamA and BamD), and LptE. The mechanism by which LptD4213 is degraded by BepA remains unclear, but its N-terminal region may interact with BepA.

## A proposed function of BepA in the promotion of the assembly of an immature LptD and LptE

BepA possibly facilitates the association of LptD with LptE on the BAM complex as the phenotypes caused by the disruption of BepA, including drug sensitivity and retarded disulfide rearrangement in LptD, can be suppressed by the overproduction of LptE (*Narita et al., 2013*). We have previously detected two species of LptD$^C$; one is not associated with LptE and observed only in an extremely early phase of its membrane assembly, and the other is associated with LptE and formed at a later phase (*Miyazaki et al., 2018*). It is likely that the LptD$^C$ molecule that was simultaneously crosslinked with BepA and BamA was not associated with LptE, as it was accumulated in the LptE-limiting condition; therefore, it would represent the former species of LptD$^C$ mentioned above. We assume that this LptD intermediate may possess a partially folded structure in which the BamA-associating C-terminal region of the barrel domain has a certain degree of a higher order (β-sheet) structure, but the BepA-associating N-terminal region is largely unstructured. Such a state of LptD may be favorable for the association of LptE with LptD (*Figure 4D*).

Interestingly, Tyr-331 of LptD that contacts with the edge-strand of BepA is located at the end of the LptE-surrounding region of the LptD β-barrel in the mature LptD/E complex (*Figure 2D*). The interaction with the edge-strand of BepA may pin the partially folded structure of LptD transiently and facilitate the association between LptD and LptE. It might also assist the formation of the β-sheet structure in the β-barrel-forming domain of LptD. Although it is possible that BepA actively promotes the LptD–LptE association by interacting with both LptD and LptE, it is also possible that BepA plays a passive role by maintaining an appropriate structure of LptD for association with LptE without showing direct interaction with LptE.

## A model of the BepA-assisted biogenesis process of LptD

The biogenesis of LptD has been well studied. LptD has been a focus of OM protein research because it provides important information on its essential cellular function and it can be used as a model for OM protein insertion into the OM by the BAM complex. Based on the results obtained in the previous and current studies, we propose a model of the BepA-assisted biogenesis of LptD (*Figure 5B*): (i) After synthesis in the cytoplasm, LptD is translocated to the periplasm through the SecYEG translocon, during or just after which the Cys residues are oxidized by DsbA to form LptD$^C$ (*Chng et al., 2012*; *Narita et al., 2013*).

It is then targeted to the BAM complex with the aid of periplasmic chaperones including SurA (*Schwalm et al., 2013*; *Vertommen et al., 2009*). (ii) The C-terminal β-signal region of LptD is inserted into the BAM complex possibly through the periplasmic ring-like structure formed by the BamA POTRA domains and the lipoprotein subunits of the BAM complex, and it establishes interaction with BamD and BamA (*Lee et al., 2019*). (iii) Although details of the exact juncture at which interaction occurs are unknown, BepA with its protease activity repressed by His-246 residue interacts with the largely unstructured N-terminal half of the LptD β-barrel-forming domain whose

C-terminal β-signal region is associated with the seam region of BamA. This stabilizes the partially unfolded assembly intermediate of LptD on the BAM complex to help the association of LptE with LptD. (iv) Then, the folding (β-sheet formation) of the unstructured N-terminal region of the LptD β-barrel domain occurs to form a premature form of LptD with a substantially folded β-barrel domain and LptE within it, like LptD4213 (*Lee et al., 2019*; *Lee et al., 2016*). (v) The β-barrel domain of LptD β-barrel is finally released from BamA and closed to form the mature LptD/E complex.

The isomerization of two disulfide bonds in LptD (LptD$^C$ to LptD$^{NC}$) should occur at a later step after the association of LptD and LptE (*Lee et al., 2016*; *Miyazaki et al., 2018*). The LptD intermediates that are stalled at certain steps in the above-mentioned processes as a result of misfolding are eliminated by the action of several peptidases including DegP (in the periplasm) and BepA /YcaL (on the BAM complex) (*Soltes et al., 2017*).

### Future perspectives of BepA study

Our study reports several new findings on the interaction of BepA with LptD and BamA on the BAM complex where BepA plays a crucial role in the biogenesis and degradation of LptD. Further, we proposed a model explaining the role of BepA in these processes. Nonetheless, there are many questions that warrant further investigation. It would be especially important to elucidate the mechanism by which the substrate gains access to the active site buried inside the BepA molecule and the manner in which the switching of BepA from the state with chaperone-like function to that with protease function occurs. The possible movement of the α6- and α9-loops should be directly examined. It is possible that there are signals that arise from the BAM complex and/or from (stalled) LptD to induce this structural/functional conversion of BepA. Further study, including structural and biochemical analysis of the BepA–LptD–BAM complex, is necessary to substantiate our model and to elucidate the molecular details of BepA functions. It remains an open question whether BepA also acts in the assembly and quality control of some other OM proteins, like in the case of LptD. Systematic identification and analysis of the additional BepA substrate OM proteins will be needed to know how generally BepA acts in biogenesis/degradation of OM proteins. Cell surface proteins are suitable drug targets as they are more easily accessible from the external milieu than cytoplasmic proteins. These studies can provide a basis for the development of new drugs targeted to BepA/LptD/BAM.

## Materials and methods

**Key resources table**

| Reagent type (species) or resource | Designation | Source or reference | Identifiers | Additional information |
|---|---|---|---|---|
| strain, strain background (*Escherichia coli*) | *E. coli* strains | This study | N/A | *Supplementary file 1* |
| strain, strain background (P1 bacteriophage) | P1vir | Laboratory stock | CGSC12133 | |
| recombinant DNA reagent | Plasmids | This study | N/A | *Supplementary file 2* |
| sequence-based reagent | PCR primers | This study | N/A | described in the below |
| antibody | Penta-His HRP conjugate(mouse monoclonal) | QIAGEN | 34460 | (1:2000 or 1:3000 dilution) |
| antibody | Anti-BepA (rabbit polyclonal) | *Narita et al., 2013* | N/A | (1:10000 dilution) |
| antibody | Anti-LptD (rabbit polyclonal) | *Narita et al., 2013* | N/A | (1:50000 dilution) |
| antibody | Anti-BamA (rabbit polyclonal) | *Gunasinghe et al., 2018* | N/A | (1:20000 dilution) |

*Continued on next page*

*Continued*

| Reagent type (species) or resource | Designation | Source or reference | Identifiers | Additional information |
|---|---|---|---|---|
| antibody | Anti-BamD (rabbit polyclonal) | *Gunasinghe et al., 2018* | N/A | (1:10000 dilution) |
| antibody | Goat Anti-Rabbit IgG (H + L)-HRP Conjugate | Bio-Rad Laboratories | 1706515 RRID:AB_2617112 | (1:5000) |
| chemical compound, drug | H-p-Bz-Phe-OH | Bachem | F2800 | |
| chemical compound, drug | Methionine, L-[$^{35}$S] Translation Grade | American Radiolabeled Chemicals | ARS 01014 | |
| chemical compound, drug | nProtein A Sepharose 4 Fast Flow | GE Healthcare | 17528004 | |
| chemical compound, drug | Ni-NTA Agarose | QIAGEN | 30250 | |
| commercial assay or kit | ECL Western Blotting Detection Reagents | GE Healthcare | RPN2106 | |
| commercial assay or kit | ECL Prime Western Blotting Detection Reagents | GE Healthcare | RPN2232 | |
| software, algorithm | Microsoft Excel | Microsoft | RRID:SCR_016137 | |
| software, algorithm | Bio-imaging Analyzer BAS-1800, BAS-5000 | Fujifilm/GE Healthcare | N/A | |
| software, algorithm | Image Qaunt LAS 4000 mini | Fujifilm/GE Healthcare | N/A | |
| software, algorithm | Multi Gauge | Fujifilm/GE Healthcare | RRID:SCR_014299 | |

## Bacterial strains and plasmids

*E. coli* K12 strains and plasmids used in this study are listed in *Supplementary files 1* and *2*, respectively. Details of the strain and plasmid construction and media are described in Construction of mutant strains and Plasmids construction, respectively.

## Media and bacterial cultures

*E. coli* cells were grown in L-rich medium (10 g/L bacto-tryptone, 5 g/L bacto-yeast extract, 5 g/L NaCl; pH adjusted to 7.2 with NaOH) or M9 synthetic medium (without CaCl₂; *Miller, 1972*) supplemented with maltose (final 0.2%), glycerol (final 0.4%), all amino acids (except Met and Cys; final concentration of 20 µg/mL each). 50 µg/mL ampicillin (Amp), 20 µg/mL chloramphenicol (Cm), 25 µg/mL kanamycin (Km), 25 µg/mL tetracycline (Tet), and 50 µg/mL spectinomycin (Spc) were added as appropriate for growing plasmid-bearing cells and selection of transformants and transductants. Bacterial growth was monitored with Mini photo 518R (660 nm; TAITEC Co., Saitama, Japan).

## Construction of mutant strains

RM2091 (JE6631, *purC80*::Tn*10*) was constructed by transferring the *purC80*::Tn*10* marker, which is located near the *bepA* gene, from CAG18470 (*Nichols et al., 1998*) into JE6631 (*Miyazaki et al., 2018*), respectively, by P1 transduction. RM2243 (AD16, *bepA(E137Q) purC80*::Tn*10*) was constructed as follows. pRM330 (a plasmid carrying *bepA(E137Q)*, see below) was introduced into RM2091 to yield cells with pRM330 integrated into the chromosome by homologous recombination in the *bepA* region. They were then grown on an L-agar plate containing 5% sucrose to select cells

that had lost the integrated plasmid. The plasmid-cured cells were screened for those having the chromosomal *bepA(E137Q)* allele at the *bepA* locus. The *bepA(E137Q)* allele was finally transferred to AD16 (*Kihara et al., 1995*) by joint P1 transduction with the *purC80*::Tn*10* marker. One of such strains was named RM2243. RM3654 (AD16, Δ*bepA*, *bamA+* *zae502*::Tn*10*) and RM3655 (AD16, *bamA(S439C)* *zae502*::Tn*10*) were constructed as follows, pRM845 (a plasmid carrying *bamA (S439C)*, see below) was introduced into YH188 (JE6631, *zae502*::Tn*10*) (*Hizukuri and Akiyama, 2012*) to yield cells with pRM845 integrated into the chromosome by homologous recombination in the *bamA* region. They were then grown on an L-agar plate containing 5% sucrose to select cells that had lost the integrated plasmid. The plasmid-cured cells were screened for those having the chromosomal *bamA(S439C)* allele at the *bamA* locus. The *bamA(S439C)* allele was transferred to SN56 (*Narita et al., 2013*) by joint P1 transduction with the *zae502*::Tn*10* marker. Strains having the *bamA+* allele and the *bamA(S439C)* allele were picked up and named RM3654 and RM3655, respectively. RM2831 (HM1742, *kan araC*-P$_{araBAD}$-*lptD*) were constructed by essentially the same procedure as the construction of strains with a chromosomal C-terminal his$_{10}$-tagged gene (*Miyazaki et al., 2020b*). First, a *kan araC*-P$_{araBAD}$-*lptD* fragment having a sequence identical to the upstream or downstream region of the *lptD* start codon at the respective ends of the fragment, was PCR-amplified from pRM741 (a plasmid carrying a *kan* cassette at the upstream of an *araC*-P$_{araBAD}$) using a pair of primers, ara-lptD-f (5′-TTGTCACGCGCAACGTTACCGATGATGGAACAATAAAATCAACGTCATA TGAATATCCTCCTTAG-3′) and ara-lptD-r (5′-GGTGGCAATCATGGTGGCCAGGAGAGTGGGGA TACGTTTTTTCATGGTGAATTCCTCCTGCTAG-3′). Then, the chromosomal *lptD* locus of the *E. coli* DY330 strain was replaced by this fragment using the λ-Red recombination system (*Yu et al., 2000*). The *kan araC*-P$_{araBAD}$-*lptD* was finally transferred to HM1742 by P1 transduction.

## Plasmids construction

pSTD689-derived plasmids carrying a *bepA* mutants were constructed from pRM290 (pSTD689-*bepA*) (*Daimon et al., 2017*) by site-directed mutagenesis. Derivatives of pRM291 (pSTD689-*bepA (E137Q)*) (*Daimon et al., 2017*) carrying an additional Cys mutation and derivatives of pUC-*bepA (E137Q)-his$_{10}$* (pUC18-*bepA(E137Q)-his$_{10}$*) (*Narita et al., 2013*) carrying an *amber* mutation were constructed by site-directed mutagenesis. pUC18-*bepA(Pro)-his$_{10}$* and pUC18-*bepA(amb)-his$_{10}$* plasmids were constructed from pUC-*bepA-his$_{10}$* (pUC18-*bepA-his$_{10}$*) (*Narita et al., 2013*) by site-directed mutagenesis. To construct pUC18-*bepA(Pro, amb, E137Q)-his$_{10}$* plasmids, a mutation for the individual Pro substitutions was introduced into each of the pUC18-*bepA(amb, E137Q)-his$_{10}$* plasmids by site-directed mutagenesis. pNB91 (pMW118-*bepA(E137Q)*) was constructed by subcloning an EcoRI-HindIII *bepA(E137Q)* fragment prepared from pUC-*bepA(E137Q)* (pUC18-*bepA (E137Q)*) (*Narita et al., 2013*) into the same sites of pMW118. pRM807 (pMW118-*bepA(A106P, E137Q)*) and pRM808 (pMW118-*bepA(F107P, E137Q)*) were also constructed by subcloning an EcoRI-HindIII *bepA* fragment from each of pSTD639-*bepA* plasmids into the same sites of pMW118.

Plasmids carrying *lptD(amb)-his$_{10}$* were constructed from pRM309 (pRM294-*lptD-his$_{10}$*) (*Miyazaki et al., 2018*) by site-directed mutagenesis. pRM821 (pRM294-*lptD(Y331amb, D749amb)-his$_{10}$*) and pRM822 (pRM294-*lptD(Y331amb, Y726amb)-his$_{10}$*) were constructed from pRM626 (pRM294-*lptD(Y331amb)-his$_{10}$*) by site-directed mutagenesis. pRM829 (pRM294-*lptD(E733C)-his$_{10}$*) and pRM831 (pRM294-*lptD(Y331, E733C)-his$_{10}$*) were constructed from pRM309 and pRM626, respectively, by site-directed mutagenesis. Plasmid carrying *lptD(Cys)-his$_{10}$* were constructed as follows. Each mutation for the Cys substitutions was introduced to pRM309 by site-directed mutagenesis. The BamHI-HindIII *lptD(Cys)-his$_{10}$* fragment of the resulting plasmids was subcloned into the same site of pTWV228 (Takara Bio Inc, Shiga, Japan) to generate pTWV228-*lptD(Cys)-his$_{10}$*.

pRM320 (pUC118-*bepA-yfgD*) was constructed by PCR amplification of the *bepA-yfgD* fragment from the genome of MC4100 using a pair of primers, bepA-f (5′-GCGCGCGGATCCATTTGAG TGGGCTAATCTTCG-3′) and yfgD-r (5′-GCGCGCGTCGACCGAACTACGCGAAGTTAATCC-3′), and, cloning it into the BamHI-SalI site of pUC118 (Takara Bio Inc) after digestion with these enzymes. For the construction of pRM324, the *bepA(E137Q)* mutation was introduced into pRM320 by site-directed mutagenesis. pRM330 (pK18mobsacB-*bepA(E137Q)-yfgD*) was constructed by subcloning the BamHI-SalI *bepA(E137Q)-yfgD* fragment from pRM324 into the same sites of pK18mobsacB (*Schäfer et al., 1994*).

pRM823 (pUC118-*bamA*) was constructed by in vitro recombination using In-Fusion HD Cloning Kit (Takara Bio Inc) of an EcoRI-BamHI fragment from pUC118 and a *bamA* fragment prepared by

PCR amplification from the genome of MC4100 using a pair of primers, bamA-f (5′-GCGCGAA TTCAGGAAGAACGCATAATAACG-3′) and bamA-r (5′-GCGCGGATCCTTACCAGGTTTTACCGATG-3′). For the construction of pRM836, the *bamA(S439C)* mutation was introduced into pRM823 by site-directed mutagenesis. pRM845 (pK18mobsacB-*bamA(S439C)*) was constructed by subcloning the EcoRI-BamHI *bamA(S439C)* fragment from pRM823 into the same sites of pK18mobsacB.

## Immunoblotting analysis

Acid-denatured proteins were solubilized in SDS-sample buffer (62.5 mM Tris-HCl (pH 6.8), 2% SDS, 10% glycerol, and 5 mg/mL bromophenol blue) with or without 10% β-ME, boiled at 98℃ for 5 min, separated by SDS-PAGE and electro-blotted onto a PVDF membrane (Merck Millipore; Billerica, MA). The membrane was first blocked with 5% skim milk in PBST (Phosphate Buffered Saline with Tween 20), and then incubated with Penta-His HRP conjugate (1:2000 or 1:3000 dilution), anti-BepA (1:10,000), anti-LptD (1:50,000), anti-BamA (1:20,000), or anti-BamD (1:10,000). After washing with PBST, the membrane was incubated with a horseradish peroxidase (HRP)-conjugated secondary antibody (1:5000) (Goat Anti-Rabbit IgG (H+L)-HRP Conjugate; Bio-Rad Laboratories, Inc, Hercules, CA) in PBST (this step was omitted for the detection using Penta-His HRP Conjugate). Proteins were visualized with ECL Western Blotting Detection Reagents (GE Healthcare UK Ltd, *Amersham Place* Little Chalfont, England) or ECL Prime Western Blotting Detection Reagents (GE Healthcare) and LAS4000 mini lumino-image analyzer (GE Healthcare).

## Pulse-chase analysis for assay of the LptD disulfide-isomerization

Cells were first grown at 30℃ in M9-medium supplemented with 2 µg/mL thiamine, 0.4% glycerol, 0.2% maltose, all amino acids (except Met and Cys) with or without 0.05% arabinose until early log phase. After induction with 1 mM IPTG for 15 min, cells were pulse-labeled with 370 kBq/mL [$^{35}$S] methionine for 1 min. At appropriate time points after addition of excess nonradioactive Met (final conc. 250 µg/mL), total cellular proteins were precipitated with 5% TCA, washed with acetone, solubilized in SDS buffer (50 mM Tris-HCl (pH 8.1), 1% SDS, and 1 mM EDTA) and diluted 33-fold with Triton buffer (50 mM Tris-HCl (pH 8.1), 150 mM NaCl, 2% Triton X-100, and 0.1 mM EDTA). After clarification, samples were incubated with anti-LptD antibodies and nProtein A Sepharose 4 Fast Flow (GE Healthcare) at 4℃ overnight with slow rotation. Proteins bound to the antibody/ProteinA-Sepharose were recovered by centrifugation, washed with Triton buffer, and then with 10 mM Tris-HCl (pH 8.1) and eluted by incubation at 98℃ for more than 5 min in SDS-sample buffer. The samples were divided into two portions and one was treated with 10% β-ME. The proteins were separated by SDS-PAGE, and visualized with BAS1800 phosphoimager. Relative LptD$^{NC}$ were calculated by the following equation: Relative LptD$^{NC}$=[LptD$^{NC}_{(X\ min)}$]/[LptD$^{C}_{(5\ min)}$+LptD$^{NC}_{(5\ min)}$], where LptD$^{NC}$ and LptD$^{C}$ are the intensities of the respective bands.

In vivo photo-crosslinking analysis, pEVOL-pBpF expresses an evolved tRNA/aminoacyl-tRNA synthetase pair that enable in vivo incorporation of *p*BPA into an *amber* codon site of a target protein by *amber* suppression. UV exposure of a cell expressing a *p*BPA-incorporated protein causes the generation of a covalent crosslinking between the *p*BPA in the target protein and a nearby protein, allowing the detection of their in vivo interaction (*Chin and Schultz, 2002*; *Miyazaki et al., 2020a*; *Young et al., 2010*). For the experiments in *Figure 1D* and *Figure 3—figure supplement 1*, cells were grown at 30℃ in L-medium containing 0.5 mM *p*BPA and 0.02% arabinose until early log phase and induced with 1 mM IPTG for 1 hr. A half volume of the cell cultures was put on a petri dish and UV-irradiated at 4℃ for 10 min using B-100AP UV lamp (365 nm; UVP, LLC, Upland, CA), at the distance of 4 cm. The other half was kept on ice as non-UV-irradiated samples. Membrane fractions were prepared by sonical cell disruption and the following ultracentrifugation, and solubilized with SDS buffer (50 mM Tris-HCl (pH 8.1), 1% SDS, and 1 mM EDTA). After 33-fold diluted with Triton buffer (50 mM Tris-HCl (pH 8.1), 150 mM NaCl, 2% Triton X-100, and 0.1 mM EDTA) and clarification, samples were subjected to pull-down with Ni-NTA Agarose (QIAGEN). The isolated proteins were solubilized in SDS-sample buffer (62.5 mM Tris-HCl (pH 6.8), 2% SDS, 10% glycerol, and 5 mg/mL bromophenol blue) containing ME, boiled at 98℃ for 5 min, and analyzed by SDS-PAGE and immunoblotting analysis.

For the experiments in *Figures 2A, B* and *3A*, cells were grown at 30℃ in L-medium containing 0.5 mM *p*BPA and 0.02% arabinose until early log phase and induced with 1 mM IPTG for 3 hr. While

a half volume of the cell cultures was UV-irradiated for 10 min, the other half was kept on ice as non-UV-irradiated samples, as above. Total cellular proteins were precipitated with 5% TCA, washed with acetone, and solubilized in SDS-sample buffer containing ME and boiled at 98°C for 5 min. The samples were subjected to SDS-PAGE and immunoblotting analysis.

For experiments in *Figure 4A and B*, cells were grown at 30°C in L-medium containing 0.5 mM *p*BPA and 0.02% arabinose until early log phase and induced with 1 mM IPTG for 3 hr. While a half volume of the cell cultures was UV-irradiated for 30 min, the other half was kept on ice as non-UV-irradiated samples, as above. Membrane fractions were prepared as above and solubilized in SDS buffer. After dilution with Triton buffer and clarification, samples were subjected to pull-down with Ni-NTA Agarose. The isolated proteins were solubilized in SDS-sample buffer containing ME, boiled at 98°C for 5 min, and analyzed by SDS-PAGE and immunoblotting analysis.

## BMB crosslinking combined with photo-crosslinking

Cells were grown at 30°C in L-medium containing 0.5 mM *p*BPA and 0.02% arabinose until early log phase and induced with 1 mM IPTG for 3 hr. BMB crosslinking was performed essentially according to the previously described procedures (*Lee et al., 2019*). After quenching the BMB crosslinking, cells were treated with or without UV-irradiation for 30 min at 4°C. Total cellular proteins were precipitated with 5% TCA, washed with acetone, and suspended in SDS-sample buffer containing 12.5 mM NEM for blocking the free Cys residues. After dilution with Triton buffer and clarification, samples were subjected to pull-down with Ni-NTA Agarose. The isolated proteins were suspended in SDS-sample buffer containing ME, boiled at 98°C for 5 min, and analyzed by SDS-PAGE and immunoblotting analysis.

## Acknowledgements

The authors thank our laboratory members and S Narita for discussion and helpful advices. This work was supported by the Japan Society for the Promotion of Science KAKENHI Grants 18H06047, 19K21179, and 20K15715 (to RM), and 15H01532 and 18H023404 (to YA), and research grants from the Institute for Fermentation Y-2020-02-027 (to RM) and from the Nagase Science and Technology Foundation (to YA).

## Additional information

### Funding

| Funder | Grant reference number | Author |
| --- | --- | --- |
| Japan Society for the Promotion of Science | 18H06047 | Ryoji Miyazaki |
| Japan Society for the Promotion of Science | 19K21179 | Ryoji Miyazaki |
| Japan Society for the Promotion of Science | 20K15715 | Ryoji Miyazaki |
| Japan Society for the Promotion of Science | 15H01532 | Yoshinori Akiyama |
| Japan Society for the Promotion of Science | 18H023404 | Yoshinori Akiyama |
| Institute for Fermentation, Osaka | Y-2020-02-027 | Ryoji Miyazaki |
| Nagase Science Technology Foundation | | Yoshinori Akiyama |

The funders had no role in study design, data collection and interpretation, or the decision to submit the work for publication.

## Author contributions
Ryoji Miyazaki, Conceptualization, Data curation, Funding acquisition, Validation, Investigation, Visualization, Methodology, Writing - original draft, Writing - review and editing; Tetsuro Watanabe, Kohei Yoshitani, Investigation; Yoshinori Akiyama, Conceptualization, Data curation, Supervision, Funding acquisition, Validation, Visualization, Methodology, Writing - original draft, Project administration, Writing - review and editing

## Author ORCIDs
Yoshinori Akiyama  https://orcid.org/0000-0003-4483-5408

## Decision letter and Author response
Decision letter https://doi.org/10.7554/eLife.70541.sa1
Author response https://doi.org/10.7554/eLife.70541.sa2

## Additional files

### Supplementary files
• Supplementary file 1. Strains used in this study.
• Supplementary file 2. Plasmids used in this study.
• Transparent reporting form

### Data availability
All data generated or analysed during this study are included in the manuscript and supporting files. Source data files have been provided for Figures 1, 2, 3, and 4, Figure 1-figure supplements 2, 3, and 4, Figure 2-figure supplements 1, and 2, Figure 3-figure supplements 1, and 2, and Figure 4-figure supplements 1, 2, and 3.

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
