## [Decision Letter]

**Acceptance summary:**

This is a wonderful study reporting a critical role of the edge-strand of the periplasmic metalloprotease BepA in the equality control of an outer membrane protein in gram-negative bacteria. This result provides new insights into the biogenesis mechanisms of outer membrane proteins and the maintenance of outer membrane integrity.

**Decision letter after peer review:**

Thank you for submitting your article "Edge-strand of BepA interacts with immature LptD on the β-barrel assembly machine to direct it to on-and off-pathways" for consideration by *eLife*. Your article has been reviewed by 3 peer reviewers, , including Heedeok Hong as the Reviewing Editor and Reviewer #1, and the evaluation has been overseen by Gisela Storz as the Senior Editor. The following individual involved in review of your submission has agreed to reveal their identity: Yihua Huang (Reviewer #2).

Essential revisions:

1. Requested experiments: the following experiments will help strengthening the conclusions.

1a) This is a suggestion that can support authors' conclusion that BepA has a chaperone-like activity. Although authors showed that A106P retains proteolytic activity, a test seems needed for other mutations. The interpretation of Figure 1 data is based on the assumption that the mutations only locally perturb the structure of BepA not severely affecting the structural integrity of other regions. Would an in vitro proteolytic assay using a structureless substrate be possible at an elevated temperature?

1b) In Figure 4c and Figure 3B, a β-ME treatment control should be added in the supporting information to confirm the crosslinking band of BepA×LptD×BamA.

2. Clarification is needed whether the chaperone-like activity facilitates LptD maturation without proteolysis, or involves proteolysis as a part of quality control mechanisms. If the latter is true, can we still say that BepA is a chaperone? Also, please, add more explanations regarding the molecular mechanisms that underlie the chaperone-like activity.

2a) Line 151-Line 160 and Figure 1D: the evidence of the pulse-chase experiments to examine the chaperone-like function of BepA is relatively weak. Since the N105P, A106P and F107P mutants disrupt the interaction with LptD and they have weaker proteolytic activities, these could result in LptDC accumulation to make the radio of LptDNC lower for these mutants. It's difficult to ignore the effects of the proteolytic activities in this assay. Please, clarify this point.

2b) I recommend that authors add an additional sentence to lines 161-169, for example, "It is unclear at this point whether the apparent chaperone-like activity is due to proteolysis or not." or clarify this point in another way to guide readers.

2c) In Discussion, it will be helpful to describe a possible molecular principle of how the binding of the edge-strand to the substrate can lead to chaperone activity.

3. Please, emphasize broader implications of this work in OMP quality control in a way that this manuscript can better fit to the scope of *eLife*. Also, the arguments for the unique findings of this work in comparison to related studies need to be strengthened further.

3a) For example, a general reader would want to know more about:

3a-i) In what other biological processes is BepA involved?

3a-ii) Is LptD the only known substrate of BepA?

3a-iii) Is the involvement of the ternary complex (i.e., BAM, an OMP client, and a quality control enzyme) a novel finding in the biogenesis of OMPs? Also, can the proposed mechanism be generalizable to OMP quality control?

3b) Reviewer 3 raises a concern that the impact of this work is incremental rather than substantial in the field of OM/OMP biogenesis. Please, provide convincing arguments on the following points:

3b-i) It is possible that the edge strand may just be reporting on substrate binding to the protease domain active site. While this may be important for substrate recognition, it does not mean that the edge strand-substrate interaction plays a deterministic role in subsequent protein triage during LptD assembly. The role of the edge strand in substrate binding has been known for other M48 metalloprotease family proteins.

3b-ii) The manuscript describes in detail how BepA likely establishes the interaction with immature LptD on the Bam complex, but many of the general conclusions into the mechanism of LptD assembly have already been described elsewhere both by this group and others. As a result, the data in the work provides limited additional insight to the general mechanism of action of BepA.

3b-iii) Figure 4 and 5: The models presented in Figures 4D, 5A, and 5B are essentially reproductions from Lee 2019, Daimon 2020, and Tomasek 2020. The main conceptual advance to the model is largely explained by Figure 4D and I'm not sure Figure 5 is warranted given its limited additional insight. I would perhaps move the entirety of Figure to the figure-supplements.

*Reviewer #1 (Recommendations for the authors):*

1. This study is highly focused on BepA-LptD-BAM although the result can have a broad impact as a mode of protein quality control mechanisms in general. Readers may be lost in the midst of the detailed descriptions of experimental systems (M48 metalloproteases, BepA, LptD, LptD4213, etc.) and experimental results.

I recommend that authors emphasize broader implications of this work in the introduction and discussion. For example, readers would want to know more about:

Is LptD the only substrate of BepA?

What other biological processes is BepA involved?

Is the involvement of the ternary complex (i.e., BAM, a OMP client, and a quality control enzyme) in LptD biogenesis a novel finding in the field?

Would the proposed mechanism be generalizable to other BepA-mediated quality control of OMPs?

2. Discussion is excellent. Nonetheless, it could be more easily conveyed to readers if it is divided into several subsections with proper headings.

3. Clarification of the LptD mutant, LptD4213, is needed (in one sentence).

*Reviewer #2 (Recommendations for the authors):*

In all, this manuscript provides novel and detailed findings regarding the interaction of BepA with LptD and BamA in the BAM complex where BepA plays a crucial role in the biogenesis and degradation of LptD. The experiment designs are ingenious; the data presented in the manuscript are elegant and generally support the conclusions. I recommend this manuscript for publication if the following issues are currently addressed.

1. Line 151-Line 160 and Figure 1D, the evidence of the pulse-chase experiments to examine the chaperone-like function of BepA is relatively weak. Since the N105P, A106P and F107P mutants disrupt the interaction with LptD and they have weaker proteolytic activities, these could result in LptDC accumulation to make the radio of LptDNC lower for these mutants. It's difficult to ignore the effects of the proteolytic activities in this assay.

2. Does BepA only degrade LptDC not LptDNC?

3. In Figure 4c and Figure 3B, a β-ME treatment control should be added to further confirm the crosslinking band of BepA×LptD×BamA.

4. Line 161-Line 165, it is not very clear why the authors performed Cys substitution for positions of N-105, A106, or F107? If they try to draw the conclusion that "the secondary structure of the edge-strand, but not the individual amino acid residues, is important for its function", they may mutate the individual position to different types of residues for experiments.

5. For Figure 1, the authors should clearly label out the catalytic residues mentioned in the main text.

6. In general, it is a little bit hard to understand how come the edge-stand of BepA is able to directly contact with its substates as this this strand (β2) appears to be buried in the crystal structure of BepA. The authors should discuss this in the Discussion part.

7. The crosslinking assay revealed that BepA can interact with an LptD intermediate assembling on BAM complex. Does the dual function of BepA depend on the BAM complex?

*Reviewer #3 (Recommendations for the authors):*

My main criticism is the conclusion that the edge strand of BepA plays an active role to direct it (LptD) to on- and off- pathways. This is possible, but it would be important to show a mutation in the edge strand that could uncouple the chaperone and proteolytic activities of BepA. In both the A106P and F107P mutants, the loss in proteolytic activity is always accompanied by a loss in chaperone activity. I recognize that identifying such a mutant is not trivial, but the title of the paper implies that the edge strand plays a larger role in the functional switch in BepA.

As is, the manuscript describes in detail how BepA likely establishes the interaction with immature LptD on the Bam complex. but many of the general conclusions into the mechanism of LptD assembly have already been described elsewhere both by this group and others. As a result, the data in the work provides limited additional insight to the general mechanism of action of BepA.

Detailed changes:

1) I would consider including a reference and an explanation describing the antibiotic sensitivity and complementation assays. Specifically, I think it would be helpful to explain how you can use erythromycin sensitivity to test BepA function in the antibiotic sensitivity assays and to explain how glucose/arabinose/IPTG regulate protein levels in the strains used for the complementation assays. These details may not be obvious to everyone and would be helpful for a more general readership. The description can be short and described in the figure captions.

2) All the structure figures look a little fuzzy and should be rendered at a higher resolution. This may be an artifact of the compression during submission, though.

3) Figure 1A (right): The 'catalytic residues' label is misspelled. The green in Regulatory His246 is also hard to read. I recommend changing the color.

4) Figure 1D: I am weary about over interpreting the quantification of relative LptDNC levels, especially with the BepA F107P variant. I'm not sure if this is the best way to characterize BepA chaperone activity and the overall conclusions of this experiment still hold even if the quantification is removed.

5) Figure 2B and 3A: LptD pBPA E391 and Q393 both change the migration pattern of LptD rather noticeably. Do you know why this is the case? BPA substitution causes some changes at other positions, but these two in particular stand out. It is almost as if there is a deletion.

6) Figure 4: Figure 4C seems to suggest that the BepA antibody may also weakly cross react with BamA. If that's the case, then it is likely that the higher of the two molecular weight adducts labeled as BepAxLptD in both Figure 4A and 4B (lane 7) are actually LptDxBamA adducts. This band should be labeled with an asterisk instead.

7) Figure 4 and 5: The models presented in Figures 4D, 5A, and 5B are essentially reproductions from Lee 2019, Daimon 2020, and Tomasek 2020. The main conceptual advance to the model is largely explained by Figure 4D and I'm not sure Figure 5 is warranted given its limited additional insight. I would perhaps move the entirety of Figure to the figure-supplements.

---

## [Author Response]

Essential revisions:1. Requested experiments: the following experiments will help strengthening the conclusions.1a) This is a suggestion that can support authors' conclusion that BepA has a chaperone-like activity. Although authors showed that A106P retains proteolytic activity, a test seems needed for other mutations. The interpretation of Figure 1 data is based on the assumption that the mutations only locally perturb the structure of BepA not severely affecting the structural integrity of other regions. Would an in vitro proteolytic assay using a structureless substrate be possible at an elevated temperature?

Thank you for the suggestion. Unfortunately, we have not succeeded in reproducibly detecting the proteolytic activity of BepA with purified BepA even when an unstructured substrate (a-casein) is used and the assay was conducted at an elevated temperature, possibly because the proteases activity of isolated BepA is tightly repressed by the mechanism that included His-246-mediated regulation as described in our paper (please see Introduction). Although BepA mutants with a mutation of His-246 or a deletion of H9 loop (these mutations release the His-246-mediated repression) significantly degrade a-casein, a combination of these mutations with the edge-strand mutations should make the interpretation of the results complicated. We thus think that the suggested experiments cannot be conducted soon.

Instead, we described the following points in the revised manuscript. Although we mentioned the self-cleavage activity of only the A106P mutant in the original manuscript, our results showed that the other edge-strand Pro mutants (other than F107P) exhibited significant self-cleavage activities as well (Figure 1—figure supplement 2B). In addition, the Pro mutants other than the A106P mutant degraded misfolded or unfolded BamA at a detectable level (Figure 1—figure supplement 2A). Furthermore, all the Pro mutants accumulated at a level comparable to that of wild-type BepA. These observations together indicate that the Pro mutations specifically affected the edge-strand structure, but not drastically altered the active site or the protein's overall structures. We described the above points in the revised text (line 174 in p7 to line 181 in p8). The differential effects of the Pro mutations on the degradation of LptD and BamA, and the self-cleavage of the C-terminal tag (Figure 1B and Figure 1—figure supplement 2A and 2B) might reflect their different interaction properties and/or affinities for the BepA edge-strand, as stated in lines 140–142 in p6.

1b) In Figure 4c and Figure 3B, a β-ME treatment control should be added in the supporting information to confirm the crosslinking band of BepA×LptD×BamA.

For Figure 4C, as the crosslinker used (BMB) is not cleavable with β-ME, we did not include a β-ME treatment control. However, the presumed crosslinked product between BamA(S430C) and LptD(E773C) was generated in a BMB- and Cys-dependent manner (Figure 4—figure supplement 2A, see below), strongly suggesting that it was indeed the BMB-mediated crosslinked product between BamA and LptD. For Figure 3B, a β-ME treatment control had been shown in the original figure (the middle panel). Please note that in this experiment, the crosslinked products were purified by making use of the His-tag attached to LptD, and detected by anti-BamA immunoblotting. Thus, no crosslinked product was detected after the β-ME treatment.

We noticed that the original Figure 4—figure supplement 2A contained several errors: aBepA should read aBamA, and LptD-His_10_xBepA should read LptD-His_10_xBamA. These errors have been corrected in new Figure 4—figure supplement 2A. We sincerely apologize to the editors and the reviewers for these errors.

2. Clarification is needed whether the chaperone-like activity facilitates LptD maturation without proteolysis, or involves proteolysis as a part of quality control mechanisms. If the latter is true, can we still say that BepA is a chaperone? Also, please, add more explanations regarding the molecular mechanisms that underlie the chaperone-like activity.2a) Line 151-Line 160 and Figure 1D: the evidence of the pulse-chase experiments to examine the chaperone-like function of BepA is relatively weak. Since the N105P, A106P and F107P mutants disrupt the interaction with LptD and they have weaker proteolytic activities, these could result in LptDC accumulation to make the radio of LptDNC lower for these mutants. It's difficult to ignore the effects of the proteolytic activities in this assay. Please, clarify this point.

Thank you for pointing out the problem of the data presentation. We re-analyzed the pulse- chase data. To examine only the chaperone-like activities (promotion of the LptD^NC^ generation) without the possible effect of the LptD degradation by the residual proteolytic activities of the A106P and the F107P mutants, the relative amounts of LptD^NC^ at each chase point to the total LptD (LptD^C^ + LptD^NC^) at the 5 min chase point were calculated and plotted as quantified data (new Figure 1D, right panel). The data clearly showed that the expression of WT BepA in the D*bepA* strain accelerated the maturation of LptD (that is, generation of LptD^NC^). Also, the A106P and the F107P mutant were mostly defective in the promotion of the LptD maturation. These results were described in the revised manuscript (lines 156–166 in p7). In addition, we previously showed that the expression of the protease-dead mutants, E137Q and H136R, significantly promoted the LptD maturation (please see Figure 4 in Narita *et al.* PNAS, *110*, E3612–E3621, 2013; doi:10.1073/pnas.1312012110) and partially suppressed the OM-defective phenotypes of the *bamB/bepA* and *bamE/bepA* double-knockout strains (Figure 6 in Narita *et al.* PNAS, *110*, E3612–E3621, 2013; doi:10.1073/pnas.1312012110) (note that in Figure 1D of the current work, the expression of the E137Q mutant only slightly promoted the LptD maturation. This would be ascribable to some difference in the experimental conditions including a lower expression level of the BepA mutant protein in the experiment for Figure 1D). These results collectively show that BepA possesses the chaperon-like activity (LptD maturation promoting activity) independent of its protease activity.

2b) I recommend that authors add an additional sentence to lines 161-169, for example, "It is unclear at this point whether the apparent chaperone-like activity is due to proteolysis or not." or clarify this point in another way to guide readers.

Thank you for the suggestion. However, as stated above, we think that our previous and current data indicated that BepA possesses the chaperone-like activity independent of its protease activity. We thus consider it inappropriate to add the suggested sentence. But, if you still think that the inclusion of the suggested sentence is essential, we are willing to do so.

2c) In Discussion, it will be helpful to describe a possible molecular principle of how the binding of the edge-strand to the substrate can lead to chaperone activity.

We suppose that the interaction of BepA (via the edge-strand) with an assembly intermediate of LptD on the BAM complex stabilizes the partially unfolded assembly intermediate of LptD on the BAM complex to help the association of LptE with LptD. Sentences explaining this point were added to Discussion (lines 388–392 in p16) and the legend to Figure 5.

3. Please, emphasize broader implications of this work in OMP quality control in a way that this manuscript can better fit to the scope of eLife. Also, the arguments for the unique findings of this work in comparison to related studies need to be strengthened further.3a) For example, a general reader would want to know more about:3a-i) In what other biological processes is BepA involved?

Our previous (Figure 8B in Daimon *et al.* Mol. Microbiol. 106, 760-776, 2017; DOI:10.1111/mmi.13844) and the current (Figure 1—figure supplement 2A in this paper) results showed that BepA can degrade misassembled (and possibly misfolded or unfolded) BamA in the D*surA* strain. Thus, BepA might also act in the proteolytic quality control of other OMPs including BamA. It remains an open question whether BepA also acts in the assembly of some other OM proteins, like in the case of LptD. Systematic identification and analysis of the additional BepA substrates will be needed to know how generally BepA acts in the biogenesis/degradation of OM proteins.

These points were described in the revised text (lines 71–73 in p3 and lines 411–412 in p16).

3a-ii) Is LptD the only known substrate of BepA?

Please see our response to (3a-i).

3a-iii) Is the involvement of the ternary complex (i.e., BAM, an OMP client, and a quality control enzyme) a novel finding in the biogenesis of OMPs? Also, can the proposed mechanism be generalizable to OMP quality control?

A ternary complex formation among an assembly intermediate of an OM protein (EspP), a periplasmic chaperone (SurA/Skp), and the BAM complex has been suggested from biochemical studies including crosslinking, although the ternary complex was not directly detected (Ieva *et al.* PNAS, E383-391, 2011; doi:10.1073/pnas.1103827108). We here experimentally demonstrated the ternary complex formation for LptD, BepA, and the BAM complex. A similar client-chaperone-BAM ternary complex might be formed in the assembly of other OM proteins. This should be addressed in a future study.

The above points were described in the revised text (lines 292–297 in p12 and lines 409–412 in p16).

3b) Reviewer 3 raises a concern that the impact of this work is incremental rather than substantial in the field of OM/OMP biogenesis. Please, provide convincing arguments on the following points:3b-i) It is possible that the edge strand may just be reporting on substrate binding to the protease domain active site. While this may be important for substrate recognition, it does not mean that the edge strand-substrate interaction plays a deterministic role in subsequent protein triage during LptD assembly. The role of the edge strand in substrate binding has been known for other M48 metalloprotease family proteins.

Our data demonstrated that the edge-strand of BepA directly binds a substrate. As pointed out by the reviewer, the involvement of the edge-strand in substrate binding has been known for other proteases. However, it was not known whether the substrate binding at the edge-strand contributes to the chaperone-like function; it was possible that the binding sites of a substrate on BepA during its proteolysis and its maturation are totally different as the chaperone-like activity of BepA is independent of its protease activity (it was conceivable, for example, that substrate binding during it maturation occurs on the surface of the C-terminal TPR domain that has been shown to interact with LptD). Our results showed that the defective binding of a substrate (LptD) at the edge-strand impairs not only its proteolysis but also its normal maturation (assembly). Because the edge-strand-bound substrate would be directly presented to the proteolytic active site for its degradation, this binding step should be important for the determination of the fates of the bound substrate. Our results strongly suggest that the substrate binding by the edge-strand is a crucial common step required for the subsequent protein triage during the LptD assembly.

3b-ii) The manuscript describes in detail how BepA likely establishes the interaction with immature LptD on the Bam complex, but many of the general conclusions into the mechanism of LptD assembly have already been described elsewhere both by this group and others. As a result, the data in the work provides limited additional insight to the general mechanism of action of BepA.

The results by us and others have previously shown that BepA has both proteolytic and chaperone-like (that is, LptD-assembly promoting) activities. Although it has been easily expected that the edge-strand-mediated interaction with a substrate is important for its degradation, it is not known how BepA executes its chaperone-like function and how BepA distinguishes the on- and off-pathway substrates. As described above, we showed that the normal interaction of the edge-strand of BepA with LptD is required not only for the degradation but also, unexpectedly, for the assembly promotion of this OMP substrate. Also, we demonstrated that BepA forms a ternary complex with LptD and the BAM complex in which BepA interacts with an unstructured N-terminal half of the LptD barrel domain and that the C-terminal region of LptD is associated with the seam strand of BamA. A ternary complex like this was first detected in this study. Based on these novel findings and the previous results, we proposed in this work the model of the BepA function in the LptD biogenesis. In this model, we suggested that the sequential movements of the a6 and a9 segments, which regulate the substrate binding and its proteolysis, respectively, enable the triage of LptD. We thus consider that our current results and the proposed model provide important clues to answer the above questions and substantial new insights into the mechanisms of the BepA functions and the LptD/OMP biogenesis. We believe that these findings and models are novel and worth publishing in a prestigious journal like *eLife*.

3b-iii) Figure 4 and 5: The models presented in Figures 4D, 5A, and 5B are essentially reproductions from Lee 2019, Daimon 2020, and Tomasek 2020. The main conceptual advance to the model is largely explained by Figure 4D and I'm not sure Figure 5 is warranted given its limited additional insight. I would perhaps move the entirety of Figure to the figure-supplements.

Figure 5A describes the possible sequential movement of the a6 and a9 segments, which will enable the triage of LptD. Figure 5B explains how the BepA-LptD-BAM ternary complex is formed during the LptD biogenesis and how the LptD intermediate is targeted to maturation or degradation. We think that these figures help readers understand our model and thus would like to keep them in the main part of the paper, if possible, although we are happy to move them to the figure-supplements if the editors and the reviewers recommend it.

Reviewer #1 (Recommendations for the authors):1. This study is highly focused on BepA-LptD-BAM although the result can have a broad impact as a mode of protein quality control mechanisms in general. Readers may be lost in the midst of the detailed descriptions of experimental systems (M48 metalloproteases, BepA, LptD, LptD4213, etc.) and experimental results.I recommend that authors emphasize broader implications of this work in the introduction and discussion. For example, readers would want to know more about:Is LptD the only substrate of BepA?

Our previous (Figure 8B in Daimon *et al.* Mol. Microbiol. 106, 760-776, 2017; DOI:10.1111/mmi.13844) and the current (Figure 1—figure supplement 2A in this paper) results showed that BepA can degrade misassembled (and possibly mis- or un-folded) BamA in the D*surA* strain. Thus, BepA might also act in the proteolytic quality control of other OMPs including BamA. It remains an open question whether BepA also acts in the assembly of some other OM proteins, like in the case of LptD. Systematic identification and analysis of the additional BepA substrates will be needed to know how generally BepA acts in the biogenesis/degradation of OM proteins.

These points were described in the revised text (lines 71-72 in p3, and lines 411–414 in p16).

What other biological processes is BepA involved?

Please see our response to the above question.

Is the involvement of the ternary complex (i.e., BAM, a OMP client, and a quality control enzyme) in LptD biogenesis a novel finding in the field?

A ternary complex formation among an assembly intermediate of an OM protein (EspP), a periplasmic chaperone (SurA/Skp), and the BAM complex has been suggested from biochemical studies including crosslinking, although the ternary complex was not directly detected (Ieva *et al.* PNAS, E383-391, 2011; doi:10.1073/pnas.1103827108). We here experimentally demonstrated the ternary complex formation for LptD, BepA, and the BAM complex. A similar client-chaperone-BAM ternary complex might be formed in the assembly of other OM proteins. This should be addressed in a future study.

The above points were described in the revised text (lines 292–297 in p12 and lines 409–412 in p16).

Would the proposed mechanism be generalizable to other BepA-mediated quality control of OMPs?

Please see our response to the above question.

2. Discussion is excellent. Nonetheless, it could be more easily conveyed to readers if it is divided into several subsections with proper headings.

Thank you for your valuable suggestion. We divided Discussion into subsections with proper headings according to the suggestion.

3. Clarification of the LptD mutant, LptD4213, is needed (in one sentence).

We added explanations of LptD4213 at two places ((1) lines 242–246 in p10, and (2) lines 353–357 in p14). Also, the properties of LptD4213 were described at other places ((3) lines 68–71 in p3, and (4) line 394 in p16). We suppose that these provide readers with sufficient information about LptD4213.

The details of the description are as follows:

1) It has been recently shown that LptD4213, a mutant form of LptD that has a short (23 amino acids) deletion in an extracellular loop (eL4) and is stalled on the BAM complex mimicking a late assembly complex (Lee et al., 2016), interacts with the seam region formed by the N- and C-terminal β-strands (β1 and β16, respectively) in the BamA barrel domain and forms a hetero-complex, in which the C-terminal β-signal of the LptD4213 was associated with the β1 strand of the BamA seam (Lee et al., 2019).

2) Notably, it has been reported that BepA expression can result in the degradation of a mutant form of LptD (LptD4213) that probably mimics a late-state assembly intermediate of LptD (Lee et al., 2019, 2016) that possesses a substantial degree of a higher order (b-barrel-like) structure and interacts with the BAM components (BamA and BamD), and LptE. The mechanism by which LptD4213 is degraded by BepA remains unclear, but its N-terminal region may interact with BepA.

3) While BepA promotes the LptD^C^ to LptD^NC^ conversion (chaperone-like function) (Narita et al., 2013), it also degrades the stalled or misassembled LptD^C^ molecules that are generated due to an *lptD* mutation (*lptD4213*) or decreased availability of or weakened interaction with LptE (protease function) (Narita et al., 2013; Soltes et al., 2017).

4) Then, the folding (b-sheet formation) of the unstructured N-terminal region of the LptD β-barrel domain occurs to form a premature form of LptD with a substantially folded β-barrel domain and LptE within it, like LptD4213 (Lee et al., 2019, 2016)

Reviewer #2 (Recommendations for the authors):1. Line 151-Line 160 and Figure 1D, the evidence of the pulse-chase experiments to examine the chaperone-like function of BepA is relatively weak. Since the N105P, A106P and F107P mutants disrupt the interaction with LptD and they have weaker proteolytic activities, these could result in LptDC accumulation to make the radio of LptDNC lower for these mutants. It's difficult to ignore the effects of the proteolytic activities in this assay.

Thank you for pointing out the problem of the data presentation. We re-analyzed the pule chase data. To examine only the chaperone-like activities (promotion of the LptD^NC^ generation) without the possible effect of the LptD degradation by the residual proteolytic activities of the A106P and the F107P, the relative amounts of LptD^NC^ at each chase point to the total LptD (LptD^C^ + LptD^NC^) at the 5 min chase point were calculated and plotted as quantified data (new Figure 1D, right panel). The data clearly showed that the expression of WT BepA in the D*bepA* strain accelerated the maturation of LptD (that is, generation of LptD^NC^). Also, the A106P and the F107P mutant were mostly defective in the promotion of the LptD maturation. These results were described in the revised manuscript (lines 156–166 in p7). In addition, we previously showed that the expression of the protease-dead mutants, E137Q and H136R, significantly promoted the LptD maturation (please see Figure 4 in Narita *et al.* PNAS, *110*, E3612–E3621, 2013; doi:10.1073/pnas.1312012110) and partially suppressed the OM-defective phenotypes of the *bamB/bepA* and *bamE/bepA* double-knockout strains (Figure 6 in Narita *et al.* PNAS, *110*, E3612–E3621, 2013; doi:10.1073/pnas.1312012110) (note that in Figure 1D of the current work, the expression of the E137Q mutant only slightly promoted the LptD maturation. This would be ascribable to some difference in the experimental conditions including a lower expression level of the BepA mutant protein in the experiment for Figure 1D). These results collectively show that BepA possesses the chaperone-like activity (LptD maturation promoting activity) independent of its protease activity.

2. Does BepA only degrade LptDC not LptDNC?

Our previous results (Daimon *et al.* PNAS, *117*, 27989-27996, 2020; DOI: https://doi.org/10.1073/pnas.2010301117) showed that BepA has little ability to degrade LptD^C^ and LptD^NC^ that are on the normal assembly pathway, but can degrade stalled or misassembled LptD^C^.

3. In Figure 4c and Figure 3B, a β-ME treatment control should be added to further confirm the crosslinking band of BepA×LptD×BamA.

For Figure 4C, as the crosslinker used (BMB) is not cleavable with β-ME, we did not include a β-ME treatment control. However, the presumed crosslinked product between BamA(S430C) and LptD(E773C) was generated in a BPB- and Cys-dependent manner (Figure 4—figure supplement 2A, see below), strongly suggesting that it was indeed the BMB-mediated crosslinked product between BamA and LptD. For Figure 3B, a β-ME treatment control has been shown in the original figure (the middle panel). Please note that in this experiment, the crosslinked products were purified by making use of the His-tag attached to LptD, and detected by anti-BepA immunoblotting. Thus, no crosslinked product was detected after the β-ME treatment.　We noticed that the original Figure 4—figure supplement 2A contained several errors: aBepA should read aBamA and LptD-His_10_xBepA should read LptD-His_10_xBamA. We sincerely apologize to the editors and the reviewers for these mistakes.

4. Line 161-Line 165, it is not very clear why the authors performed Cys substitution for positions of N-105, A106, or F107? If they try to draw the conclusion that "the secondary structure of the edge-strand, but not the individual amino acid residues, is important for its function", they may mutate the individual position to different types of residues for experiments.

We chose a Cys substitution, because (i) the Cys mutants can be used in the disulfide crosslinking experiments, and (ii) a previous study strongly suggested that a Cys mutation does not affect the secondary structure of an edge-strand in another protease RseP (Akiyama *et al. eLife*, e08928, 2015; DOI:10.7554/*eLife*.08928). This was described in the revised text (lines 169–172 in p7).

5. For Figure 1, the authors should clearly label out the catalytic residues mentioned in the main text.

We annotated the catalytic residues in new Figure 1A to increase the clarity of the figure.

6. In general, it is a little bit hard to understand how come the edge-stand of BepA is able to directly contact with its substates as this this strand (β2) appears to be buried in the crystal structure of BepA. The authors should discuss this in the Discussion part.

We suppose that in both assembly and degradation pathways of LptD, an unknown signal(s), such as a specific interaction of BepA with the substrate and/or the BAM complex, induces a structural change in BepA (dislocation of the a6 loop covering the active site) to expose its active site and to enable the interaction between the LptD polypeptide and the edge-strand, whereas His-246 continues to repress the degradation of LptD by BepA. This was described in Discussion (lines 337–341 in p14). Please also see the model of Figure 5A.

7. The crosslinking assay revealed that BepA can interact with an LptD intermediate assembling on BAM complex. Does the dual function of BepA depend on the BAM complex?

We now have no direct evidence but think it possible that the BAM complex is involved in the switching of the dual functions of BepA through its direct interaction with BepA. This point was discussed in Discussion (lines 337–341 and 344 to 348 in p14, and lines 408–409). We would like to investigate this possibility in a future study.

Reviewer #3 (Recommendations for the authors):My main criticism is the conclusion that the edge strand of BepA plays an active role to direct it (LptD) to on- and off- pathways. This is possible, but it would be important to show a mutation in the edge strand that could uncouple the chaperone and proteolytic activities of BepA. In both the A106P and F107P mutants, the loss in proteolytic activity is always accompanied by a loss in chaperone activity. I recognize that identifying such a mutant is not trivial, but the title of the paper implies that the edge strand plays a larger role in the functional switch in BepA.As is, the manuscript describes in detail how BepA likely establishes the interaction with immature LptD on the Bam complex. but many of the general conclusions into the mechanism of LptD assembly have already been described elsewhere both by this group and others. As a result, the data in the work provides limited additional insight to the general mechanism of action of BepA.

The results by us and others have previously showed that BepA has both proteolytic and chaperone-like (that is, LptD-assembly promoting) activities. Although it has been easily expected that the edge-strand-mediated interaction with a substrate is important for its degradation, it is not known how BepA executes its chaperone-like function and how BepA distinguish the on- and off-pathway substrates. As described above, we showed that the normal interaction of the edge-strand of BepA with LptD is required not only for the degradation but also, unexpectedly, for the assembly promotion of this OMP substrate. Also, we demonstrated that BepA forms a ternary complex with LptD and the BAM complex in which BepA interacts with an unstructured N-terminal half of the LptD barrel domain and that the C-terminal region of LptD is associated with the seam strand of BamA. The ternary complex like this was first detected in this study. Based on these novel findings and the previous results, we proposed in this work the model of the BepA function in the LptD biogenesis. In this model, we suggested that the sequential movements of the a6 and a9 segments, which regulates the substrate binding and its proteolysis, respectively, enables the triage of LptD. We thus consider that our current results and the proposed models provide important clues to answer the above questions and substantial new insights into the mechanisms of the BepA functions and the LptD/OMP biogenesis. We believe that these findings and model are novel and worth publishing in a prestigious journal like *eLife*.

Detailed changes:1) I would consider including a reference and an explanation describing the antibiotic sensitivity and complementation assays. Specifically, I think it would be helpful to explain how you can use erythromycin sensitivity to test BepA function in the antibiotic sensitivity assays and to explain how glucose/arabinose/IPTG regulate protein levels in the strains used for the complementation assays. These details may not be obvious to everyone and would be helpful for a more general readership. The description can be short and described in the figure captions.

According to the suggestions by the reviewer, we added brief explanations to the legends to Figure 1—figure supplement 3, Figure 2—figure supplement 2, and Figure 3—figure supplement 2 for how we can use erythromycin sensitivity to test the BepA function in the antibiotic sensitivity assays and for how glucose/arabinose/IPTG regulate protein levels in the strains used for the complementation assays.

2) All the structure figures look a little fuzzy and should be rendered at a higher resolution. This may be an artifact of the compression during submission, though.

We will upload high-resolution figures with the revised text.

3) Figure 1A (right): The 'catalytic residues' label is misspelled. The green in Regulatory His246 is also hard to read. I recommend changing the color.

We are sorry for the misspelling. We corrected the error. We also modified the color of His-246 to increase the clarity of Figurer 1A.

4) Figure 1D: I am weary about over interpreting the quantification of relative LptDNC levels, especially with the BepA F107P variant. I'm not sure if this is the best way to characterize BepA chaperone activity and the overall conclusions of this experiment still hold even if the quantification is removed.

Thank you for pointing out the problem of the data presentation. We re-analyzed the pule chase data. To examine only the chaperone-like activities (promotion of the LptD^NC^ generation) without the possible effect of the LptD degradation by the residual proteolytic activities of the A106P and the F1107P, the relative amounts of LptD^NC^ at each chase point to the total LptD (LptD^C^ + LptD^NC^) at the 5 min chase point were calculated and plotted as quantified data (new Figure 1D, right panel). The data clearly showed that the expression of WT BepA in the D*bepA* strain accolated the maturation of LptD (that is, generation of LptD^NC^). Also, the A106P and the F107P mutant were mostly defective in the promotion of the LptD maturation. These results were described in the revised manuscript (lines 156–166 in p7). In addition, we previously showed that the expression of the protease-dead mutants, E137Q and H136R, significantly promoted the LptD maturation (please see Figure 4 in Narita *et al.* PNAS, *110*, E3612–E3621, 2013; doi:10.1073/pnas.1312012110) and partially suppressed the OM-defective phenotypes of *bamB/bepA* and *bamE/bepA* double-knockout strains (Figure 6 in Narita *et al.* PNAS, *110*, E3612–E3621, 2013; doi:10.1073/pnas.1312012110) (note that in Figure 1D of this paper, the expression of the E137Q mutant only slightly promoted that LptD maturation, possibly because of difference in the experimental condition including a lower expression level of the BepA mutant protein in this experiment). These results collectively show that BepA possesses the chaperone-like activity (LptD maturation promoting activity) independent of its protease activity.

5) Figure 2B and 3A: LptD pBPA E391 and Q393 both change the migration pattern of LptD rather noticeably. Do you know why this is the case? BPA substitution causes some changes at other positions, but these two in particular stand out. It is almost as if there is a deletion.

We confirmed that the genes for LptD *p*BPA E391 and Q393 have no deletion in their coding regions by sequencing the plasmids. We now do not know the exact reason why these proteins exhibited altered mobility, but we often encounter anomalous SDS gel mobility of *p*BPA-incorporated proteins and other missense mutant proteins. The anomalous mobility of the mutant proteins could be ascribed to changes in charges, and/or local structures that are formed even in the presence of SDS.

6) Figure 4: Figure 4C seems to suggest that the BepA antibody may also weakly cross react with BamA. If that's the case, then it is likely that the higher of the two molecular weight adducts labeled as BepAxLptD in both Figure 4A and 4B (lane 7) are actually LptDxBamA adducts. This band should be labeled with an asterisk instead.

As pointed out by the reviewer, our data (Figure 4C) indicate that the BepA antibody may also weakly cross-react with BamA. However, we think that the higher of the two molecular weight adducts labeled as BepAxLptD in Figure 4A and 4B are unlikely to be the LptDxBamA adducts for the following reason; while the anti-BamA immunoblotting in Figure 4A showed that the band intensities of the LptDxBamA adducts generated with LptD(D749*p*BPA) and LptD(Y331*p*BPA/D749*p*BPA) were much stronger than that generated with LptD(Y331*p*BPA), a band whose intensity was stronger than the higher of the two adducts generated with LptD(Y331*p*BPA) was not detected for LptD(D749*p*BPA) and LptD(Y331*p*BPA/D749*p*BPA) in ani-BepA immunoblotting. Instead, a band marked with an asterisk in the anti-BepA blots (in new Figure 4) possibly indicates an LptD-BamA crosslinked product that was detected due to the apparent cross-reactivity of the anti-BepA antibody with the LptDxBamA crosslinked product. We modified Figure 4 and explained this point in the legend to Figure 4.

7) Figure 4 and 5: The models presented in Figures 4D, 5A, and 5B are essentially reproductions from Lee 2019, Daimon 2020, and Tomasek 2020. The main conceptual advance to the model is largely explained by Figure 4D and I'm not sure Figure 5 is warranted given its limited additional insight. I would perhaps move the entirety of Figure to the figure-supplements.

Figure 5A describes the possible sequential movement of the a6 and a9 segments, which will enable the triage of LptD. Figure 5B explains how the BepA-LptD-BAM ternary complex is formed during the LptD biogenesis and how the LptD intermediate is targeted to maturation or degradation. We think that these figures help readers understand our model and thus would like to keep them in the main part of the paper, if possible, although we are happy to move them to the figure-supplements if the editors and the reviewers recommend it.